# Functional canonical RNAi in mice expressing a truncated Dicer isoform and long dsRNA

Valeria Buccheri [1], Josef Pasulka[1], Radek Malik [1], Zuzana Loubalova [1,4], Eliska Taborska [1], Filip Horvat[1,2], Marcos Iuri Roos Kulmann[1], Irena Jenickova [3], Jan Prochazka [3], Radislav Sedlacek[3] & Petr Svoboda [1]✉

## Abstract

**Canonical RNA interference (RNAi) is sequence-specific mRNA degradation guided by small interfering RNAs (siRNAs) made by RNase III Dicer from long double-stranded RNA (dsRNA). RNAi roles include gene regulation, antiviral immunity or defense against transposable elements. In mammals, RNAi is constrained by Dicer's adaptation to produce another small RNA class—microRNAs. However, a truncated Dicer isoform (ΔHEL1) supporting RNAi exists in mouse oocytes. A homozygous mutation to express only the truncated ΔHEL1 variant causes dysregulation of microRNAs and perinatal lethality in mice. Here, we report the phenotype and canonical RNAi activity in $Dicer^{\Delta HEL1/wt}$ mice, which are viable, show minimal miRNome changes, but their endogenous siRNA levels are an order of magnitude higher. We show that siRNA production in vivo is limited by available dsRNA, but not by Protein kinase R, a dsRNA sensor of innate immunity. dsRNA expression from a transgene yields sufficient siRNA levels to induce efficient RNAi in heart and muscle. $Dicer^{\Delta HEL1/wt}$ mice with enhanced canonical RNAi offer a platform for examining potential and limits of mammalian RNAi in vivo.**

**Keywords** Dicer; dsRNA; Mirtron; siRNA; PKR
**Subject Category** RNA Biology

## Introduction

RNase III Dicer generates small RNAs, which guide ribonucleoprotein complexes in microRNA (miRNA) and RNA interference (RNAi) pathways (reviewed in (Ketting, 2011)). In the gene-regulating miRNA pathway, Dicer cleaves genome-encoded small hairpin precursors (pre-miRNA) into ~21–23 nt duplexes. One of the strands becomes a mature miRNA bound to an AGO protein and guides gene repression, the other strand is discarded. In RNAi, which functions in gene regulations, retrotransposon repression, and antiviral immunity, long double-stranded RNA (dsRNA) is cut by Dicer into 22 nt small interfering RNA (siRNA) duplexes. One strand is loaded onto an endonucleolytic AGO protein and guides sequence-specific RNA cleavage.

Co-existence of small RNA biogenesis and RNA-induced silencing complex (RISC) formation in RNAi and miRNA pathways evolved in different ways. In *D. melanogaster*, the pathways diverged such that each employs dedicated Dicer and AGO proteins (Lee et al, 2004). *C. elegans* utilizes a single Dicer for both pathways and small RNAs are sorted onto different AGO proteins (Ketting et al, 2001). Invertebrate RNAi provides defense against parasitic nucleic acids—this role appears minimized in vertebrates where genomic, phylogenetic, and structural data support the primary role of Dicer in the miRNA pathway (reviewed in (Zapletal et al, 2023)). At the same time, canonical RNAi (i.e., long dsRNA-induced sequence-specific mRNA degradation (Fire et al, 1998)) was occasionally observed in mammalian somatic cells (Diallo et al, 2003; Elbashir et al, 2001; Gan et al, 2002; Shinagawa and Ishii, 2003; Tran et al, 2004; Yi et al, 2003) suggesting that RNAi might exist under favorable conditions. Key constraints of mammalian endogenous canonical RNAi include (I) availability of long dsRNA, (II) inefficient long dsRNA processing by Dicer, and (III) innate immunity mechanisms, which respond to dsRNA and restrict RNAi (Demeter et al, 2019; Flemr et al, 2013; Takahashi et al, 2018; van der Veen et al, 2018). The inhibition may be reciprocal (Gurung et al, 2021; Seo et al, 2013) but there is no simple antagonistic relationship among RNAi and other dsRNA-responding pathways (Meng et al, 2013; Montavon et al, 2021).

An exceptional instance of mammalian RNAi evolved in mouse oocyte where an N-terminally truncated oocyte-specific Dicer variant (Dicer[O]) supports highly active and essential endogenous RNAi (Flemr et al, 2013). Dicer's N-terminal DExD/H helicase domain (HEL1 helicase subdomain) provides functional adaptation of the mammalian Dicer to miRNA biogenesis and inhibits processing of long dsRNA into siRNAs (Zapletal et al, 2022). Removal of HEL1 enhances long dsRNA processing in vivo and supports both, miRNA biogenesis and endogenous RNAi (Flemr et al, 2013). Mouse oocytes generate dsRNAs giving rise to siRNAs

[1]Institute of Molecular Genetics of the Czech Academy of Sciences, Videnska 1083, 142 20, Prague 4, Czech Republic. [2]Bioinformatics Group, Division of Molecular Biology, Department of Biology, Faculty of Science, University of Zagreb, 10000 Zagreb, Croatia. [3]Czech Centre for Phenogenomics, Institute of Molecular Genetics of the Czech Academy of Sciences, Prumyslova 595, 252 50 Vestec, Czech Republic. [4]Present address: National Institutes of Diabetes and Digestive and Kidney Diseases, National Institutes of Health, Bethesda, MD 20892, USA. ✉E-mail: svobodap@img.cas.cz

targeting mRNAs and retrotransposons (Tam et al, 2008; Watanabe et al, 2008) while having reduced innate immunity response to dsRNA, such as minimal levels of Protein kinase R (PKR, official gene symbol *Eif2ak2*) (Stein et al, 2005).

To activate canonical RNAi in mouse soma, we produced a mouse mutant where the endogenous *Dicer* gene was genetically engineered to express the more active Dicer variant lacking the HEL1 domain (Dicer$^{\Delta HEL1}$) (Zapletal et al, 2022). As a control, we generated an allele denoted *Dicer$^{SOM}$*, which expresses an HA-tagged full-length Dicer variant but lacks the same intronic sequences like the *Dicer$^{\Delta HEL1}$* allele (Fig. 1A) (Taborska et al, 2019). Both engineered *Dicer* alleles express Dicer proteins of expected sizes and at comparable levels (Zapletal et al, 2022). *Dicer$^{SOM/SOM}$* mice are viable but females are sterile because this modification eliminates expression of Dicer$^O$ in oocytes, which is essential for female fertility (Flemr et al, 2013; Taborska et al, 2019). *Dicer$^{\Delta HEL1/\Delta HEL1}$* mice exhibit severe miRNome dysregulation, growth retardation, defects in the cardiopulmonary system, and perinatal lethality (Zapletal et al, 2022). While the lethal phenotype of *Dicer$^{\Delta HEL1/\Delta HEL1}$* animals precludes having Dicer activity in somatic tissues from biallelic expression, *Dicer$^{\Delta HEL1/wt}$* heterozygotes are viable and fertile, demonstrating that a single *Dicer$^{\Delta HEL1}$* allele is well tolerated in vivo.

Here, we investigated the phenotype of *Dicer$^{\Delta HEL1/wt}$* mice and assessed their ability to mount functional RNAi response to long dsRNA. We report that a single *Dicer$^{\Delta HEL1}$* allele affects canonical miRNA biogenesis much less than expected while it increases biogenesis of several non-canonical miRNAs called mirtrons (Ladewig et al, 2012; Westholm and Lai, 2011). Despite siRNA biogenesis is enhanced by an order of magnitude, robust endogenous RNAi is rarely observed in organs of *Dicer$^{\Delta HEL1/wt}$* mice, even in the *Pkr* null background. Consequently, RNAi effect was observed only when transgene-based overexpression of long dsRNA in heart and skeletal muscle yielded sufficient siRNA abundance.

# Results

## *Dicer$^{\Delta HEL1/wt}$* mice have a subtle but discernible phenotype

*Dicer$^{\Delta HEL1/\Delta HEL1}$* mice were growth retarded, had developmental defects, and died perinatally (Zapletal et al, 2022). In contrast, *Dicer$^{\Delta HEL1/wt}$* animals appeared normal. Fertility of *Dicer$^{\Delta HEL1/wt}$* was normal in ICR and C57Bl/6NCrl genetic backgrounds and there was no significant difference in the birth rate and survival of wild types and heterozygotes (Fig. 1B). For detailed phenotype assessment, a phenotype screening of *Dicer$^{\Delta HEL1/wt}$* animals was performed at the Czech Centre for Phenogenomics (the full report is provided in Source Data for EV1). The phenotyping pipeline included assessment of embryonic development, anatomy, histopathology, metabolism, hematology, immunology, cardiopulmonary function, and neurobiology (including vision, hearing, and behavior). *Dicer$^{\Delta HEL1/wt}$* animals were smaller than wild-type littermates (Figs. 1C and EV1B), otherwise were anatomically normal (Source Data for EV1). In contrast to *Dicer$^{\Delta HEL1/\Delta HEL1}$* mice (Zapletal et al, 2022), heart and lung functions were normal (Fig. EV1C and Source Data for EV1). Hematopoietic parameters were also normal, including those, which were affected in *Dicer$^{\Delta HEL1/\Delta HEL1}$* mutants (Fig. EV1D). Taken together, *Dicer$^{\Delta HEL1/wt}$* animals were viable, fertile and exhibited a minor growth reduction as the major phenotype.

## Mirtrons dominate miRNome changes in *Dicer$^{\Delta HEL1/wt}$* mutants

The *Dicer$^{\Delta HEL1/\Delta HEL1}$* genotype was associated with massive miRNA dysregulation, which was consistent with altered function of Dicer lacking the N-terminal HEL1 domain (Zapletal et al, 2022). Because *Dicer$^{\Delta HEL1/\Delta HEL1}$* animals died perinatally, we analyzed small RNAs in

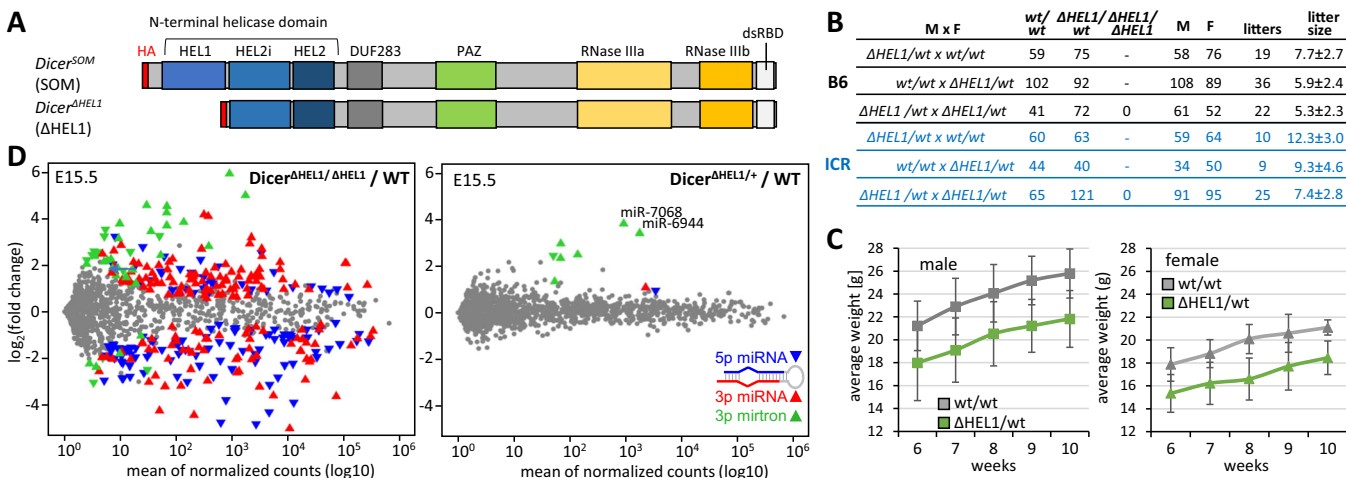

**Figure 1. Extended analysis of *Dicer$^{\Delta HEL1}$* mutants.**

(A) Schematic depiction of Dicer$^{SOM}$ and Dicer$^{\Delta HEL1}$ protein variants. (B) Breeding performance of heterozygous mutants. A table with calculated percentages for genotypes and sexes is provided as Fig. EV1A. (C) Weight of 6–10-week-old *Dicer$^{\Delta HEL1/wt}$* and wild-type littermates on the C57Bl/6NCrl background. At least 11 mice with each genotype and sex were analyzed. Error bars = SD. (D) MA plots of small RNA-seq analysis of whole *Dicer$^{\Delta HEL1/\Delta HEL1}$* and *Dicer$^{\Delta HEL1/wt}$* E15.5 embryos compared to wild type (WT) E15.5 embryos (n = 3 for each genotype). Depicted are changes in levels of annotated murine miRNAs (miRBase 22.1 (Kozomara et al, 2019)). Significantly dysregulated 5p and 3p miRNAs (DESeq p-value 0.05) are shown as oriented blue ▼ and red ▲ triangles, respectively. Significantly dysregulated mirtrons are represented by green triangles whose orientation indicates 5p and 3p miRNA strands. The published MA plot for *Dicer$^{\Delta HEL1/\Delta HEL1}$* (Zapletal et al, 2022) is included here for convenient comparison of changes in homo- and heterozygotes. Source data are available online for this figure.

E15.5 embryos where we could compare miRNome changes in *Dicer^{ΔHEL1}* homozygotes and heterozygotes. Consistent with the minor phenotype, *Dicer^{ΔHEL1/wt}* E15.5 embryos showed minimal miRNome changes when compared to wild-type littermates (Fig. 1D, right panel). Majority of significantly dysregulated miRNAs (7/9 miRNAs) were mirtrons, miRNAs with non-canonical biogenesis arising from small spliced introns (Berezikov et al, 2007). Increased levels of a mirtron subset were consistent with previously reported increased cleavage of their precursors by the Dicer^{ΔHEL1} variant (Zapletal et al, 2022).

Further analysis of miRNome changes in adult organs was done in animals that were also used for siRNA analyses described later. Thus, in addition to the *Dicer^{ΔHEL1/wt}* or control *Dicer^{SOM/wt}* genotypes, these mice were heterozygous for a *Pkr* mutation and carried a dsRNA-expressing transgene Tg(CAG-EGFP-MosIR) (Nejepinska et al, 2012), for simplicity referred to as MosIR transgene. We obtained small RNA profiles from brain, heart, liver, spleen, and thymus of *Dicer^{ΔHEL1/wt} Pkr^{+/−}* Tg(CAG-EGFP-MosIR) animals and compared them with small RNA profiles of normal age-matched C57Bl/6NCrl mice. In adult organs of *Dicer^{ΔHEL1/wt}* mice, we observed a similar picture as in E15.5 *Dicer^{ΔHEL1/wt}* embryos: canonical miRNAs showed small changes while mirtrons were among the most upregulated miRNAs (Figs. 2A and EV2). Notably, when compared to E15.5 embryos (Fig. 1D), more canonical miRNAs

appeared slightly, but significantly, differentially expressed (Fig. 2A). This was likely caused by other experimental parameters than the loss of the helicase domain. First, E15.5 embryos came from sibling females mated at the same time and were collected at the same moment, while wild-type organs were collected separately from similarly old animals (Table EV2). Second, additional genetic modifications, particularly the Tg(CAG-EGFP-MosIR) transgene, which expressed dsRNA processed by Dicer, could also have some effect on organ miRnomes. When we compared miRNAs in organs from animals, which genetically differed only by the loss of helicase in one Dicer allele (*Dicer^{ΔHEL1/wt} Pkr^{+/−}* Tg(CAG-EGFP-MosIR) vs. *Dicer^{SOM/wt} Pkr^{+/−}* Tg(CAG-EGFP-MosIR)), differential miRNA expression as well as miRNA expression variability were strongly reduced, especially in heart and spleen where mirtrons remained over-represented (Fig. 2B,C).

## siRNA biogenesis is stimulated in vivo by *Dicer^{ΔHEL1}* but not by the loss of PKR

Next, we investigated changes in siRNA production in *Dicer^{ΔHEL1/wt}* cells. Analysis of 21–23 nt RNAs made from transiently expressed long hairpin dsRNA in *Dicer^{ΔHEL1/ΔHEL1}* ESCs showed that replacement of the full-length Dicer with *Dicer^{ΔHEL1}* variant increases dsRNA-derived 21–23 nt small RNAs by an order of magnitude (Fig. 3A). Notably, eliminating PKR in *Dicer^{ΔHEL1/ΔHEL1}* ESCs added another

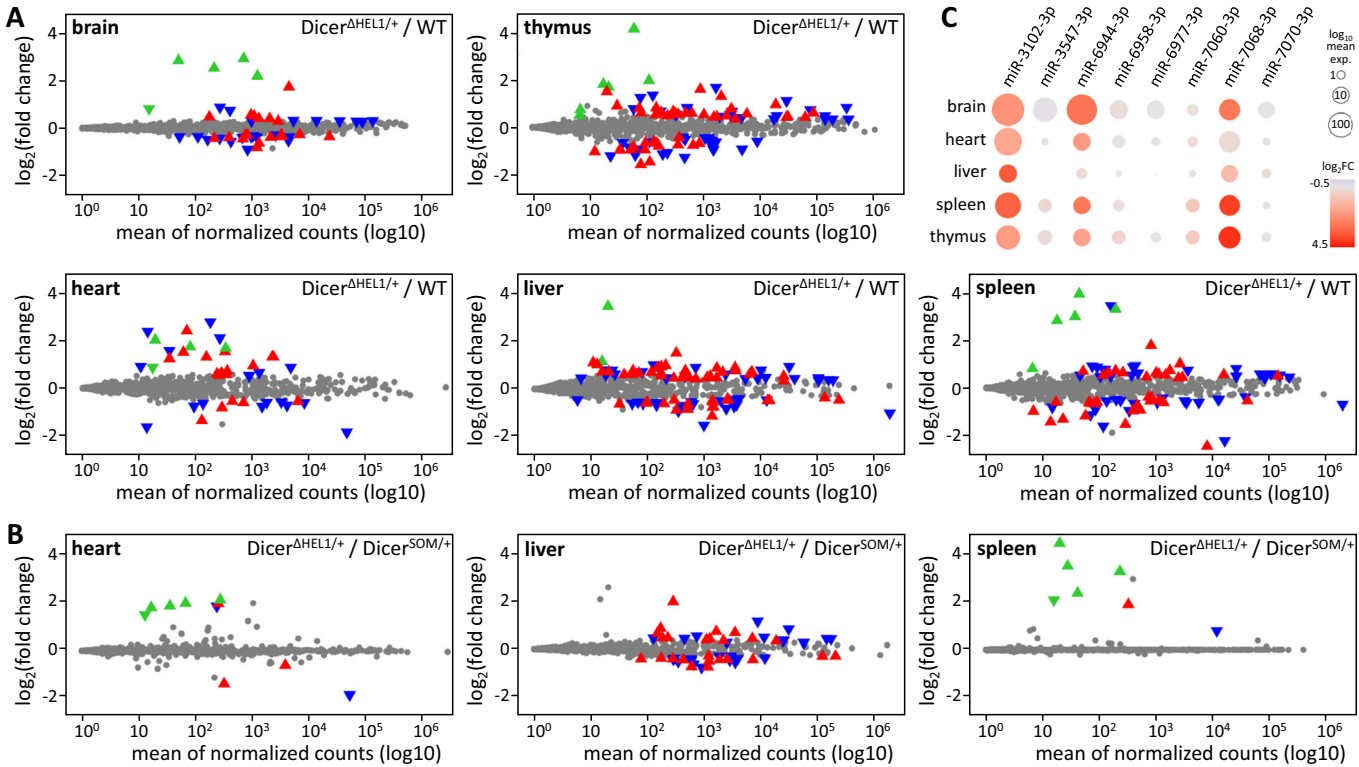

**Figure 2. miRNA dysregulation in organs of *Dicer^{ΔHEL1/wt}* mice.**

(A) MA plots depicting changes of annotated murine miRNAs (miRBase 22.1 (Kozomara et al, 2019)) in *Dicer^{ΔHEL1/wt} Pkr^{+/−}* Tg(CAG-EGFP-MosIR) animals relatively to age-matched normal C57Bl/6NCrl controls (n = 3 for each genotype). Significantly dysregulated 5p and 3p miRNAs (DESeq p-value 0.05) are shown as oriented blue ▼ and red ▲ triangles, respectively. Mirtrons are represented by green triangles whose orientation indicates 5p and 3p miRNA strands. (B) miRNA changes in *Dicer^{ΔHEL1/wt} Pkr^{+/−}* Tg(CAG-EGFP-MosIR) animals relatively to age-matched *Dicer^{SOM/wt} Pkr^{+/−}* Tg(CAG-EGFP-MosIR) animals (n = 3 for each genotype). (C) Abundance and expression changes of selected mirtrons in different organs of *Dicer^{ΔHEL1/wt} Pkr^{+/−}* Tg(CAG-EGFP-MosIR) animals relative to C57Bl/6NCrl mice.

order of magnitude in siRNA abundance (Fig. 3A). With this siRNA abundance (~6500 21–23 nt reads per million (RPM) of 19–32 nt reads), we observed sequence-specific knock-down of a luciferase reporter carrying a complementary sequence (Fig. 3B). We also observed a similar effect of PKR loss in experiments in NIH 3T3 cells while overexpression of PKR reduced RNAi in ESCs and NIH 3T3 cells (Demeter et al, 2019). Since *Pkr* knockout (*Pkr*$^{-/-}$) mice are viable and phenotypically indistinguishable from wild-type animals (Yang et al, 1995), *Pkr* knock-out seemed to offer a chance to further stimulate RNAi in our in vivo model and prompted us to produce a *Pkr* knock-out and add it to our analysis.

To evaluate siRNA production in *Dicer*$^{\Delta HEL1/wt}$ mutants, we first analyzed 21–23 nt RNAs from *Optn* and *Anks3* loci, which give rise to 21–23 nt RNAs in Dicer-dependent manner, i.e., bona fide endo-siRNAs (Flemr et al, 2013; Tam et al, 2008; Watanabe et al, 2008). The *Optn* locus carries an inverted repeat at the 3' end of the gene, which is predicted to form a stem loop with ~300 bp stem and ~1 kb loop and gives rise to low-abundant 21–23 nt small RNAs from the base-pairing region (Fig. 3C). RNA sequencing data from wild type mouse organs show that low levels of mRNA and endo-siRNAs from the *Optn* exist in many organs (Figs. 3D,E and EV3A). Most endo-siRNAs are found in muscle and heart, notably much more than in intestine, where *Optn* mRNA levels are comparable. *Optn* endo-siRNAs were strongly upregulated in tissues of *Dicer*$^{\Delta HEL1/wt}$ animals (Fig. 4A). *Anks3*, which harbors an inverted repeat in an intron, showed minimal expression in organs while upregulation of *Anks3* endo-siRNAs was observed in heart, spleen, and thymus of *Dicer*$^{\Delta HEL1/wt}$ *Pkr*$^{+/-}$ Tg(CAG-EGFP-MosIR) animals

(Fig. 4B). Remarkably, analysis of heart and liver from *Dicer*$^{\Delta HEL1/wt}$ and *Dicer*$^{SOM/wt}$ mice showed that a single allele of *Dicer*$^{\Delta HEL1}$ increased abundance of *Optn* endo-siRNAs by an order of magnitude but loss of PKR did not have a statistically significant effect (Fig. 3F, *p*-value > 0.05, one-tailed t-test).

Negligible amounts of naturally arising endo-siRNAs (Figs. 3E and 4) were consistent with our earlier analyses of Dicer$^O$ and Dicer$^{\Delta HEL1}$ variants (Demeter et al, 2019; Flemr et al, 2013) and implied that long dsRNA abundance may be the limiting factor. We thus investigated changes of endogenous siRNAs as well as siRNAs produced from dsRNA expressed from the MosIR transgene (Fig. 5A). This transgene is ubiquitously expressing a long dsRNA hairpin with *Mos* gene sequence placed in the 3' UTR of an EGFP reporter mRNA, an architecture analogous to that of *Optn* dsRNA. MosIR long dsRNA was well tolerated and able to induce RNAi in oocytes but not in a set of examined somatic organs (Nejepinska et al, 2012). The MosIR transgene has a strong synthetic promoter (CAG (Okabe et al, 1997)), which yields different levels of transgene expression in different organs (Fig. 5B,C) consistent with previously observed CAG-EGFP transgene expression (Nejepinska et al, 2012; Tchorz et al, 2012). At the same time, this allows for investigating capacity of siRNA production and efficient induction of RNAi at different MosIR expression levels.

Analysis of small RNAs from *Dicer*$^{SOM/wt}$ and *Dicer*$^{\Delta HEL1/wt}$ animals carrying the MosIR transgene showed strong increase of MosIR-derived endo-siRNA in animals carrying a single *Dicer*$^{\Delta HEL1}$ allele (Fig. 5D). MosIR siRNA abundance in *Dicer*$^{\Delta HEL1/wt}$ organs ranged over three orders of magnitude and reached ~10,000 RPM in heart

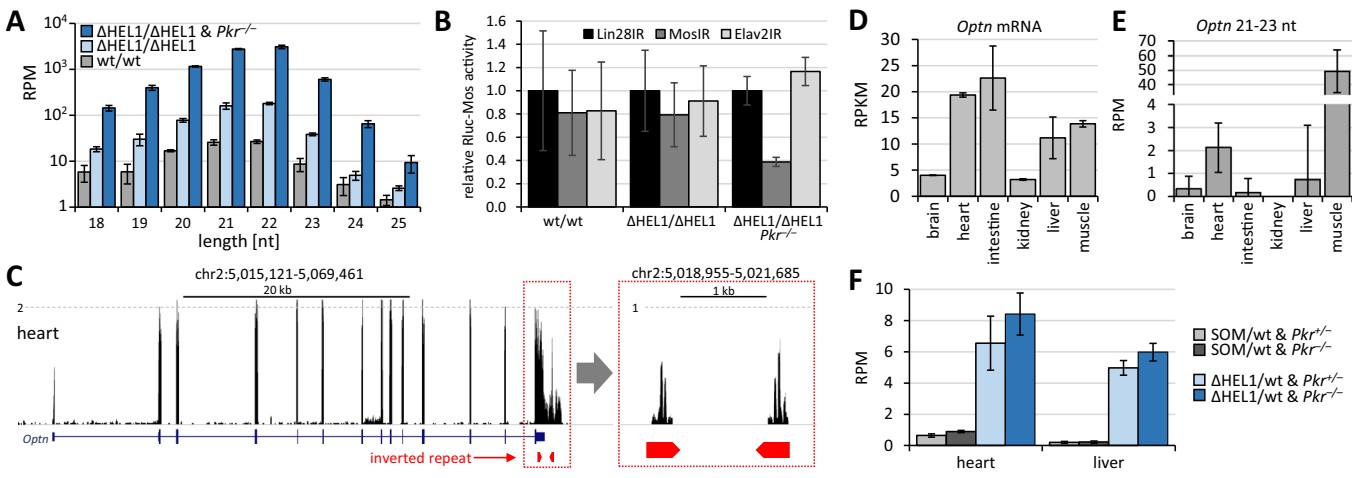

**Figure 3. Analysis of siRNA production by ΔHEL1 Dicer isoform.**

(A) In ESCs, the loss of PKR enhances production of endo-siRNAs from a dsRNA-expressing plasmid MosIR (Flemr et al, 2013). The MosIR plasmid was transfected into ESCs with indicated genotypes in triplicates in which small RNAs were analyzed by RNA-seq. The y-axis depicts reads per million (RPM) calculated for all 18–32 nt mapped reads. (B) Endogenous RNAi activity in genetically modified ESCs. RNAi was analyzed using luciferase-based RNAi assay (Demeter et al, 2019). Cells were co-transfected with plasmids expressing firefly luciferase (non-targeted reporter), *Renilla* luciferase reporter with *Mos* sequence in the 3' UTR (RNAi-targeted reporter), and one of the plasmids expressing a long hairpin RNA (Lin28IR, MosIR or Elavl2IR). Results are presented as a ratio of a *Renilla* luciferase activity divided by a non-targeted firefly luciferase activity scaled to Lin28IR, which was set to 1. Two or more transfection experiments were performed in a triplicate for each genotype and inverted repeat plasmid. (C–F) EndosiRNAs from a natural inverted repeat at the 3' end of *Optn* gene. (C) USCS browser snapshot of the *Optn* locus displaying mRNA expression in heart using published RNA-seq data (Sollner et al, 2017). Next to it are displayed 21–23 nt RNAs from small RNA sequencing data from heart of a normal mouse (Isakova et al, 2020) mapped into the *Optn* inverted repeat (depicted by red pentagons). The vertical scale is in counts per million of 18–32 nt reads (CPM). (D) Quantification of *Optn* mRNA expression in organs based on a published dataset (Sollner et al, 2017). The vertical scale is in reads per kilobase per million (RPKM), n = 3. (E) Quantification of *Optn* endo-siRNAs in the same set of organs as in the panel (D) using a published RNA-seq dataset, n = 24–28 (Isakova et al, 2020). (F) Effect of ΔHEL1 and *Pkr* deletion on *Optn* endo-siRNA abundance in heart and liver from three animals for each genotype. All bar graphs depict mean +/− SD error bars. Source data are available online for this figure.

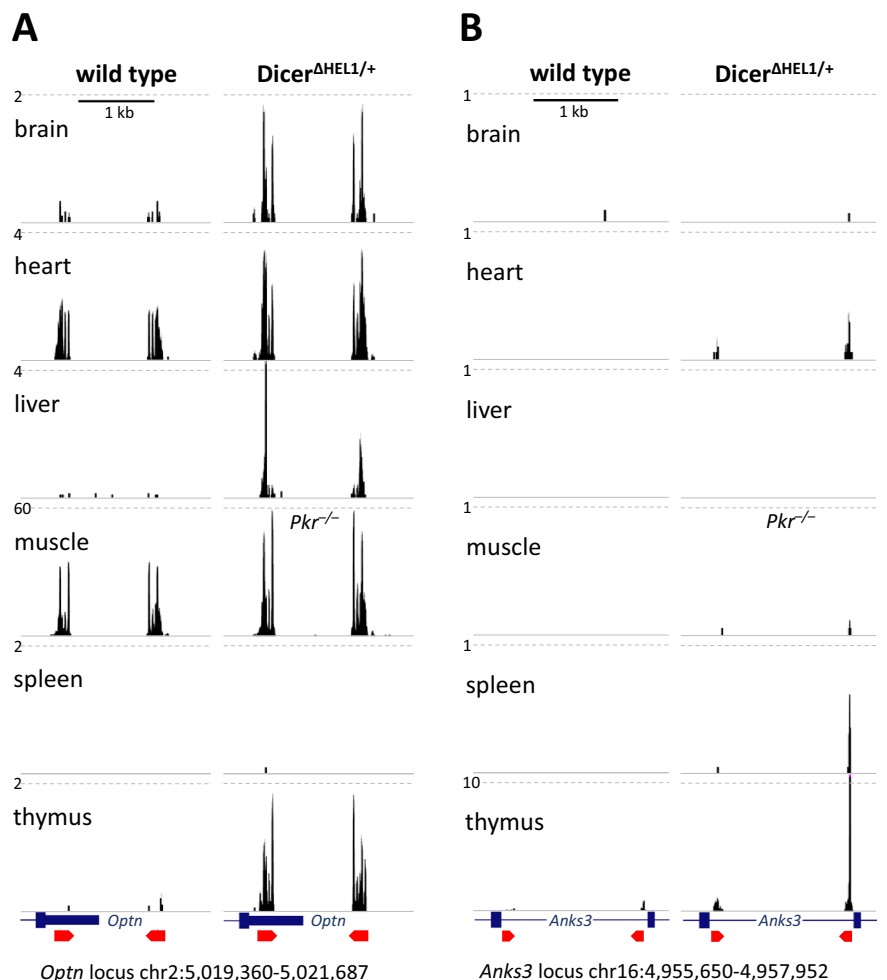

**Figure 4. Low levels of small RNAs originating from inverted repeats in *Optn* and *Anks3* loci in different organs in normal animals and their increase in ΔHEL1 mutants.**

(A) UCSC browser snapshots show 21–23 nt reads in *Optn* locus in normal and ΔHEL1 samples (*Dicer*^ΔHEL1/wt^ *Pkr*^+/-^ Tg(CAG-EGFP-MosIR) animals, except of muscle where was available only *Pkr*^-/-^ background). (B) Analogous UCSC browser snapshot for the *Anks3* locus.

and ~35,000 RPM in muscle (Fig. 6F), organs with the highest MosIR expression (Fig. 5B,C). This shows that siRNA biogenesis in *Dicer*^ΔHEL1/wt^ is primarily limited by dsRNA abundance and achieving high siRNA levels from endogenously expressed dsRNA requires high dsRNA expression (Fig. 5D) relative to endogenous levels of *Hprt* (Fig. EV3B).

Similarly to *Optn* endo-siRNAs, we did not observe any significant increase in *Mos* siRNAs in the absence of PKR in any of the analyzed organs (Fig. 5E). These data imply that the inhibitory effect of PKR on siRNA biogenesis concerns transient transfections of MosIR into cells expressing Dicer^ΔHEL1^. Absence of increased siRNA levels in organs from *Dicer*^ΔHEL1/wt^ *Pkr*^-/-^ Tg(CAG-EGFP-MosIR) animals argues against interference of PKR with endogenous RNAi in vivo. These results (Figs. 3F and 5E) prompted us to reconsider our assumption that PKR is suppressing RNAi in mammalian cells, which stemmed from cell culture experiments (Nejepinska et al, 2014). Since PKR was also reported to inhibit expression of transiently transfected plasmids (Nejepinska et al, 2014; Terenzi et al, 1999), we investigated transient transfection of MosIR-

expressing plasmids into *Dicer*^ΔHEL1/ΔHEL1^ cell lines with and without *Pkr*. We have found that cells lacking *Pkr* have higher expression of MosIR (Fig. 5F) suggesting that increased MosIR siRNA levels observed in the absence of PKR (Fig. 3A) could be just an effect of increased expression of transiently transfected MosIR-expressing plasmids. This would also explain the lack of a comparable effect in transgenic animals and would be consistent with increased levels of 24 and 25 nt MosIR fragments in ESC transfection, which were likely degradation products of the more expressed MosIR transcript (Fig. 3A).

## Dicer^ΔHEL1^ can generate high siRNA levels needed for functional RNAi in vivo

To test whether *Dicer*^ΔHEL1^ can support endogenous RNAi, we used MosIR as a source of dsRNA and, since *Mos* is not expressed in somatic tissues, we produced a transgenic mouse line carrying an mCherry-Mos reporter whose expression was also controlled by the CAG promoter Tg(CAG-mCherry-Mos). This reporter should be

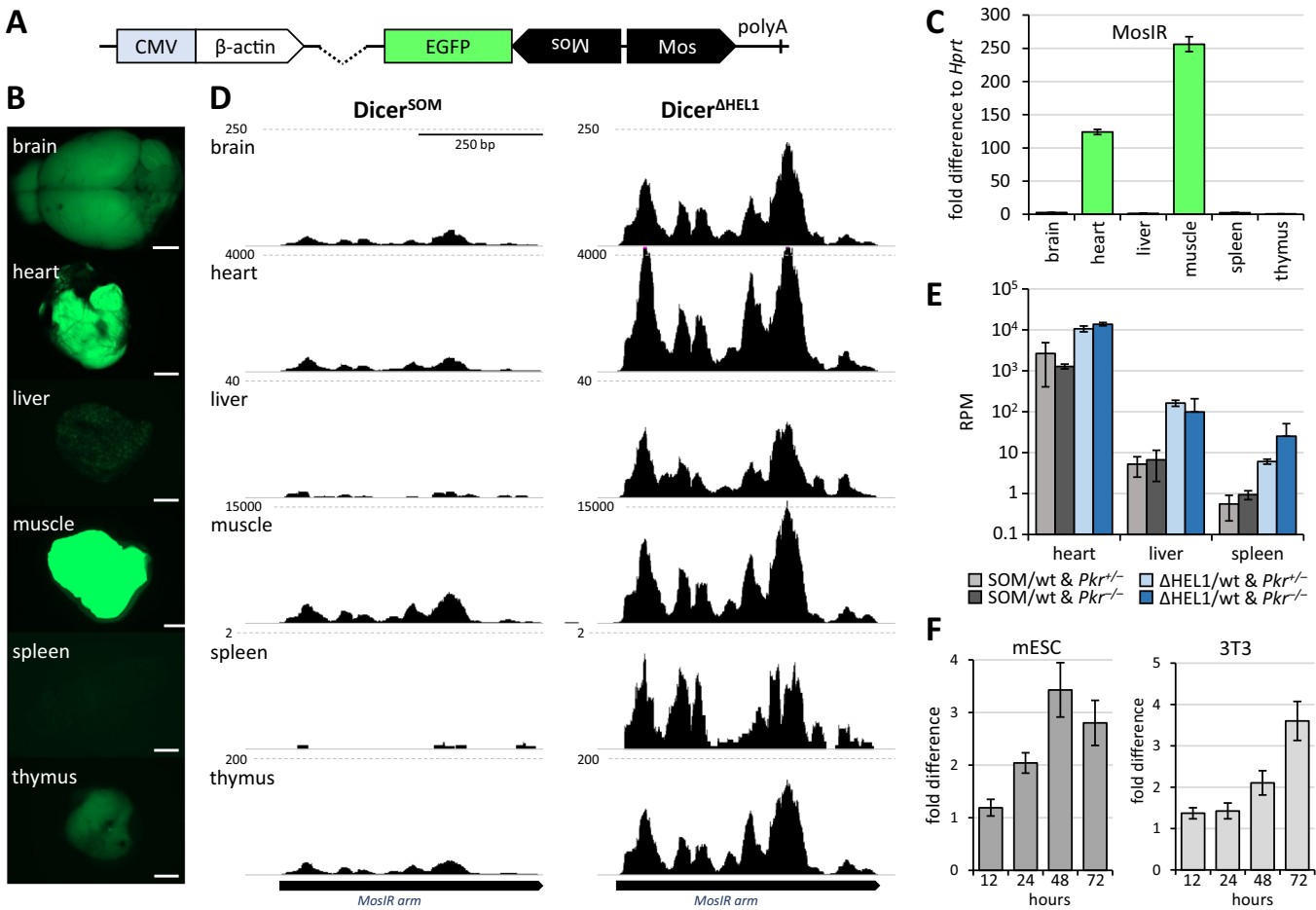

**Figure 5. MosIR-derived small RNA production in SOM and ΔHEL1 heterozygous animals.**

(A) Schematic depiction of the CAG-EGFP-MosIR transgene, a mouse transgenic line carrying this transgene has been established and analyzed previously (Nejepinska et al, 2012). (B) EGFP fluorescence in animals carrying MosIR transgene. All organs come from a single MosIR *Pkr⁻/⁻* animal. Size bar = 2 mm. (C) qPCR analysis of CAG-EGFP-MosIR transgene expression in *Dicer^{SOM/wt} Pkr⁻/⁻* organs. Expression is presented as fold difference relative to *Hprt*. For $\log_2$ relative difference to *Hprt* expression, which resolves better expression closer to that of *Hprt*, see Fig. EV4A. Data come from three (muscle and spleen) or five (other organs) biological replicates analyzed in technical triplicates. (D) Abundance and distribution of 21–23 nt RNAs derived from MosIR and mapped onto endogenous Mos sequence in different organs. All samples in the panel had *Pkr⁻/⁻* background, $n = 3$ animals for each genotype. (E) Loss of PKR does not significantly enhance production of 21–23 nt RNAs from MosIR. $n = 3$ animals for each genotype. (F) qPCR analysis of MosIR transcript abundance in *Dicer^{ΔHEL1/ΔHEL1}* ESCs and 3T3 cells lacking PKR relatively to parental *Dicer^{ΔHEL1/ΔHEL1}* cells. ESC *Dicer^{ΔHEL1/ΔHEL1}* cell lines were the same as in Fig. 1E. Data come from three experiments performed in triplicates. All bar graphs depict mean +/− SD error bars. Source data are available online for this figure.

sensitive to targeting by MosIR-derived endo-siRNAs (Fig. 6A) and has a similar pattern of expression as MosIR but had lower expression level (Figs. 6B and EV4A,B). The ratio of dsRNA to its target should be similar in different organs because MosIR and mCherry-Mos rely on the same promoter (compare Fig. 5C with 6B and EV4A with EV4B). To investigate RNAi effects, we crossed mice to combine MosIR and mCherry-Mos transgenes with *Dicer^{ΔHEL1}* or *Dicer^{SOM}* alleles and analyzed levels of the mCherry-Mos reporter in *Dicer^{ΔHEL1/wt}* or *Dicer^{SOM/wt}* mice in the presence or absence of MosIR. The analysis was done in the *Pkr⁻/⁻* background to avoid any possible interference with RNAi albeit PKR did not appear to significantly affect siRNA abundance in examined organs (Figs. 3F and 5E).

Remarkably, mCherry fluorescence was reduced by the presence of MosIR in heart and muscle of *Dicer^{ΔHEL1/wt}* mice (Fig. 6C and D, respectively) and in muscle of *Dicer^{SOM/wt}* mice (Fig. 6D). Because

fluorescence signal from the mCherry-Mos reporter in organs was not suitable for quantifying effects of MosIR, qPCR was used to estimate mCherry-Mos reporter suppression by RNAi in brain, heart, liver, muscle, spleen, and thymus. We found that MosIR-induced mCherry-Mos reporter knock-down was present only in heart and muscle of *Dicer^{ΔHEL1/wt}* animals and in muscle of *Dicer^{SOM/wt}* animals (Fig. 6E). RNAi effect thus occurred in organs where MosIR siRNA abundance reached close to $10^4$ RPM or more (Fig. 6F). mCherry-Mos reporter mRNA was knocked down in heart by 53% and 94% in muscle of *Dicer^{ΔHEL1/wt}* animals (Fig. 6E). In the context of mammalian small RNA biology, >$10^4$ RPM abundance in our RNA-seq results corresponds to the level of expression of most abundant miRNAs. Notably, MosIR siRNA abundance around 500 RPM observed in brain and thymus of *Dicer^{ΔHEL1/wt}* mice (Fig. EV4C) was insufficient to cause a detectable reporter knockdown when the reporter was expressed similarly to (brain) or even several fold less

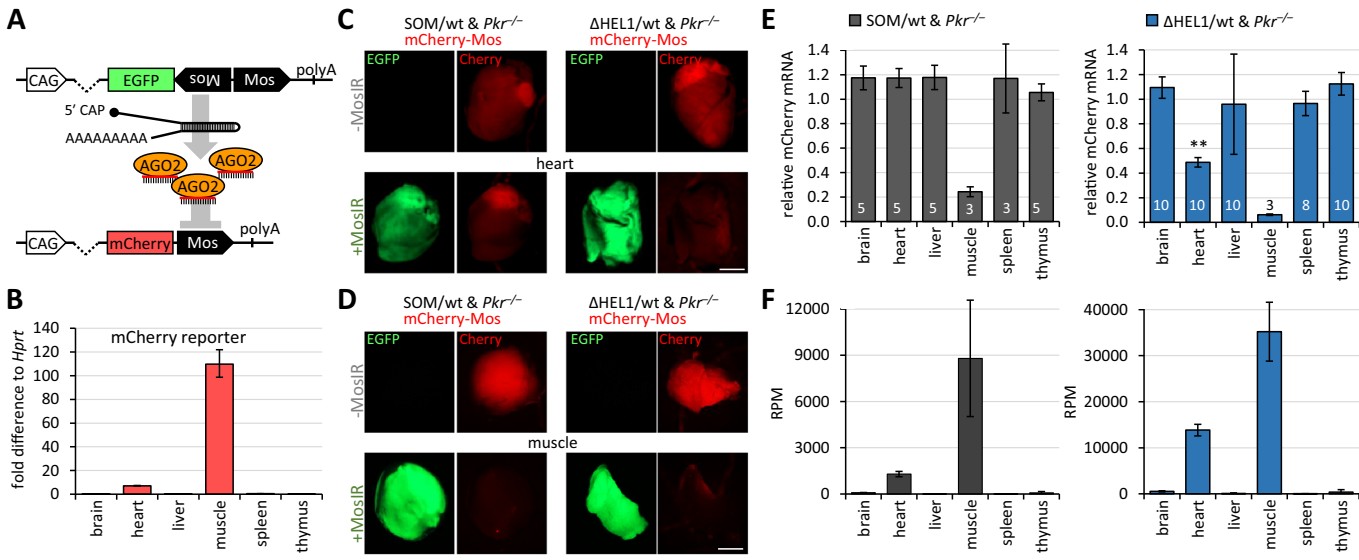

**Figure 6. RNAi induction in SOM and ΔHEL1 heterozygotes.**

(A) A scheme of the transgenic RNAi system using the transgene Tg(CAG-EGFP-MosIR) as a source of dsRNA and the transgene Tg(CAG-mCherry-Mos) as a sequence-specific target. (B) qPCR analysis of CAG-mCherry-Mos transgene expression in *Dicer^SOM/wt^ Pkr^-/-^* organs of a single animal in technical triplicates. Expression is presented as fold difference relative to *Hprt*. For log₂ relative difference to *Hprt* expression, which resolves better expression closer to that of Hprt, see Fig. EV4B. Data come from three (muscle and spleen) or five (other organs) biological replicates analyzed in technical triplicates. (C, D) Fluorescence stereomicroscopy images of hearts (C) and muscles (D) in different genotypes showing EGFP and mCherry fluorescence. Scale bar 2 mm (E) qPCR analysis of mCherry-Mos levels relative to control samples without the MosIR transgene. White numbers indicate the number of organ pairs used for comparison. ** indicates statistically significant reduction of mCherry reporter mRNA level in ΔHEL1 heterozygotes relative to SOM heterozygotes (two-tailed t-test p-value = 0.0001). (F) MosIR endo-siRNA abundance—shown is 21–23 nt RNA abundance in reads per million (RPM) of all 18–32 nt mapped small RNA reads, n = 3 animals for each genotype. Logscale visualization is available in Fig. EV4C. All bar graphs depict mean +/− SD error bars. Source data are available online for this figure.

(thymus) than *Hprt* (Fig. EV4B). Particularly absence of RNAi in thymus where the reporter was much less expressed than in brain is informative regarding the siRNA abundance threshold for achieving functional RNAi in vivo.

## Discussion

Functional and structural analyses of Dicer showed that the N-terminal helicase domain of mammalian Dicers is the critical structural element for adaptation of the enzyme for efficient miRNA biogenesis and inefficient dsRNA cleavage for canonical RNAi (Aderounmu et al, 2023; Liu et al, 2018; Ma et al, 2008; Zapletal et al, 2022). Inverse situation exists in mouse oocytes, which have highly active RNAi and inefficient miRNA pathway (Kataruka et al, 2020; Ma et al, 2010; Murchison et al, 2007; Suh et al, 2010; Tam et al, 2008; Tang et al, 2007; Watanabe et al, 2015). The key Dicer adaptation supporting highly active RNAi in mouse oocytes is a simple N-terminal truncation stemming from evolutionary emergence of an alternative oocyte-specific promoter downstream of exons encoding the HEL1 domain (Flemr et al, 2013). Mammals apparently retain functional canonical RNAi mechanism but effective target suppression requires overcoming main constraints of RNAi: available dsRNA substrate and its efficient processing into siRNAs. We have previously investigated these conditions in transient transfection in cultured cells (Demeter et al, 2019). In this work, we investigated conditions yielding functional canonical RNAi in vivo using genetically engineered mice. Our first intervention

towards more efficient RNAi was modification of endogenous *Dicer* gene to ubiquitously produce the truncated Dicer variant lacking HEL1 domain. However, *Dicer^ΔHEL1/ΔHEL1^* mice showed that the truncated variant cannot functionally substitute full-length Dicer because it lacks fidelity necessary for miRNA biogenesis (Zapletal et al, 2022). Therefore, analysis of canonical RNAi employed *Dicer^ΔHEL1/+^* mice, which provide a viable and fertile mammalian model with enhanced ability to produce siRNAs in somatic cells. Of note is that co-expression of both Dicer variants at different ratio occurs in mouse oocytes as well (Franke et al, 2017).

Ubiquitous expression of the truncated Dicer isoform in soma affects fitness. *Dicer^ΔHEL1/wt^* animals are smaller. *Dicer^ΔHEL1/ΔHEL1^* embryos exhibited major growth retardation between E12.5 and E15.5 and a slight deviation was also observed in *Dicer^ΔHEL1/wt^* animals at that time (Zapletal et al, 2022). Small RNA analyses suggest that the phenotype is associated with miRNA changes. The most pronounced miRNome change in *Dicer^ΔHEL1/wt^* E15.5 embryos and adult organs was upregulation of several mirtrons whose precursors have extended stems and would thus be suboptimal full-length Dicer substrates (the extended pre-miR-3102 stem even carries a miRNA tandem (Fig. EV2)). These mirtrons are usually low expressed in wild-type tissues but upregulated mirtrons miR-7068-3p (19-fold) and mir-6944-3p (18-fold) achieved considerable abundance in *Dicer^ΔHEL1/wt^* E15.5 embryos and could contribute to the reduced growth phenotype. *Dicer^ΔHEL1/wt^* animals thus provide an interesting model for future analysis of mirtrons significance in vivo.

An unexpected result was that heterozygosity of *Dicer^ΔHEL1^* mutation affected canonical miRNA biogenesis much less than

expected (Figs. 1D and 2B). The difference between $Dicer^{\Delta HEL1/\Delta HEL1}$ and $Dicer^{\Delta HEL1/wt}$ E15.5 embryos is striking (Fig. 1D) and suggests that the full-length Dicer contributes to canonical miRNA biogenesis significantly more than the truncated variant. We hypothesize that the pre-dicing state formed by the full-length mammalian Dicer (Liu et al, 2018; Zapletal et al, 2022) facilitates more efficient recognition and selection of canonical miRNA precursors in vivo while the $Dicer^{\Delta HEL1}$ variant is more promiscuous and thus more likely to bind and cleave suboptimal substrates of the full-length Dicer.

As expected, $Dicer^{\Delta HEL1/wt}$ mutants showed increased production of endo-siRNAs across tissues but their levels were generally negligible to induce effective target suppression. siRNA concentrations used in knock-down experiments in cultured cells (5–50 nM) would correspond to $10^4$–$10^5$ siRNA molecules per a fibroblast cell (Kataruka et al, 2020). Abundant miRNAs are in a similar range—it was estimated that a HeLa cell contains ~50,000 let-7 miRNA molecules (Bosson et al, 2014). Let-7 family miRNA abundance in our RNA-seq data from wild type somatic organs ranges between 50,000–150,000 RPM, which contrasts with $Optn$ endo-siRNAs being <10 RPM in most organs and highest ~50 RPM in muscle.

While availability of dsRNA is an obvious limiting factor for endogenous RNAi, analysis of MosIR transgene in $Dicer^{SOM/wt}$ and $Dicer^{\Delta HEL1/wt}$ mice brought interesting observations. First, sole over-expression of MosIR (which mimics $Optn$ dsRNA architecture) was sufficient to induce efficient RNAi even with endogenous full-length Dicer. The highest siRNA level produced by full-length Dicer in vivo upon dsRNA overexpression was ~9000 RPM in muscle (MosIR in $Dicer^{SOM/wt}$ mice, Fig. 6F) and induced 76% knock-down of a complementary transcript (Fig. 6E). This shows that RNAi is not absent in mammalian cells but usually terribly inefficient. Second, mice tolerate massive overexpression of long hairpin RNA in muscle and heart, where qPCR shows that MosIR transcripts are two orders of magnitude above $Hprt$ mRNA abundance. While the MosIR transgene has its expression pattern determined by the CAG promoter and we remain uncertain how high dsRNA expression might be tolerated in other organs, it nonetheless supports an idea that mammalian effective canonical RNAi appears under exceptional conditions. Third, the upper end of the wide range of Mos siRNA abundance in different organs suggests that overexpressed dsRNA hardly saturates Dicer in vivo.

In cultured cells, we observed that PKR has negative effect on RNAi—overexpression of PKR reduced RNAi in ESCs and NIH 3T3 cells while PKR knock-out had a positive effects in cell culture experiments (Demeter et al, 2019). In $Dicer^{\Delta HEL1/\Delta HEL1}$ ESCs transiently transfected with plasmids expressing long hairpin RNA and a complementary target, we observed an order of magnitude higher siRNA abundance from the inverted repeat and efficient target suppression in the absence of PKR (Fig. 3A,B). In addition, PKR binds MosIR dsRNA (Nejepinska et al, 2014) and Dicer interacts with PKR (Montavon et al, 2021). All these results lead us to assumption that loss of PKR in vivo could provide an additional boost for RNAi activity. However, in vivo data and additional analysis of cell culture experiment suggest that PKR in transient transfections reduces plasmid expression and RNAi stimulation in the absence of PKR is largely caused by higher dsRNA expression. While we cannot exclude PKR could have some inhibitory effects on RNAi in vivo, comparison of $Pkr^{+/-}$ and $Pkr^{-/-}$

genotypes suggests that PKR absence does not significantly impact RNAi effects.

Apart from Dicer activity and dsRNA availability, additional factors may influence RNAi efficiency. As mammalian RNAi employs the molecular mechanism primarily dedicated to the miRNA pathway, RNAi efficiency may be affected via the RISC effector complex. First, siRNAs also act as miRNAs (Doench et al, 2003) and a siRNA guiding a RISC is able to bind and repress partially complementary mRNAs like a miRNA (Jackson et al, 2006). Consequently, all accessible seed sequences in the transcriptome may act as a sponge for all partially complementary siRNAs and reduce probability of targeting perfectly complementary targets. At the same time, RNAi requires only a small RNA bound to AGO2 (holoRISC) while the miRNA-like activity requires the fully assembled RISC. Thus, RNAi may be more efficient when holoRISC abundance is increased. This could be achieved, for example, by disrupting interaction of AGO2 with the adaptor protein TNRC6, which provides a landing platform for additional miRNA-guided RISC components (Danner et al, 2017). Second, highly complementary targets trigger target-directed miRNA degradation (TDMD) (de la Mata et al, 2015), which operates through destabilization of AGO (Han et al, 2020; Shi et al, 2020). TDMD would reduce efficiency of RNAi by limiting multiple turnover activity of holoRISC.

Taken together, our results provide an important framework for future research on mammalian canonical RNAi. We demonstrate that induction of functional canonical RNAi in vivo in mouse soma is possible because mice can tolerate both, increased Dicer activity and high expression of long dsRNA substrate necessary to achieve efficient target knock-down. At the same time, our results show that canonical RNAi in vivo requires siRNA abundance comparable to that of abundant cellular miRNAs, presumably because canonical RNAi exploiting molecular mechanism primarily serving the miRNA pathway. Consequently, occurrence of functional mammalian RNAi would be expected to be accompanied by distinct conditions (and adaptations), which overcome its aforementioned common constrains. Such adaptations bringing together necessary amounts of Dicer activity and dsRNA can be observed in mouse oocytes, which are a notable example of functional and biologically relevant mammalian RNAi. Oocytes express $Dicer^O$ and express stable lncRNAs carrying antisense pseudogenes base pair with mRNAs from parental genes and resulting dsRNAs are efficiently processed by Dicer into siRNAs, which are able to significantly reduce specific transcript levels (Karlic et al, 2017; Tam et al, 2008; Watanabe et al, 2008).

A distinct case may be contribution of RNAi to mammalian antiviral immunity, which has been reported (Li et al, 2013; Maillard et al, 2013) but remains unclear. A study suggested that tissue stem cells produce AviD Dicer variant, which is protecting them against viruses (Poirier et al, 2021). Notably, authors could conveniently use our Dicer models to test their hypothesis in vivo as both $Dicer^{\Delta HEL1}$ and $Dicer^{SOM}$ alleles prevent alternative splicing giving rise to AviD. Importantly, antiviral RNAi would have two additional features, which experimental design standing on MosIR and a targeted reporter lacks. First, Dicer-mediated processing of dsRNA of replicating viruses could have an antiviral effect on its own. Second, we studied RNAi in a system of steady-state expression of the trigger and target. This is a different scenario than targeting a replicating

system where even a few percent reduction of its replication efficiency may have well-detectable effect after several replication cycles.

# Methods

## Animals

Tg(CAG-EGFP-MosIR), Dicer$^{SOM}$ and Dicer$^{\Delta HEL1}$ mice were produced as previously described (Nejepinska et al, 2012; Taborska et al, 2019; Zapletal et al, 2022). The original mutation was introduced into a 129 strain ESC line, which gave rise to a chimeric male mice, which was mated with ICR females to obtain germline transmission. Once the line was established, breeding was maintained on ICR background as well mice were bred onto the C57Bl/6NCrl background for at least six generations. Tg(CAG-mCherry-Mos) and Pkr (Eif2ak2) mutant mice were produced for this study and their production is described further below. Animals were housed at institute's specified pathogen-free animal facility and were fed ad libitum. Animal experiments were carried out in accordance with the Czech law and were approved by the Institutional Animal Use and Care Committee (approval no. 34-2014).

### Genotyping

For genotyping, tail biopsies were lysed in DEP-25 DNA Extraction buffer (Top-Bio) according to the manufacturer's instructions. 1 μl aliquot was used with HighQu DNA polymerase master mix for PCR. Genotyping primers are provided in Table EV1.

### Embryo and organ harvest

Mice were mated overnight, and the presence of a vaginal plug indicated embryonic day (E) 0.5. Mice were sacrificed by cervical dislocation. The embryos were collected at E15.5, washed in PBS and used for RNA sequencing. Organs collected from sacrificed animals were either directly used for analysis or stored in −80 °C for later use.

### Tg(CAG-mCherry-Mos) mouse line

To generate the Tg(CAG-mCherry-Mos) mouse line, a PCR-amplified 750 bp fragment of the N-terminal part of the Mos transcript (corresponding to nucleotides 292–1025 of the Mos cDNA sequence ENSMUST00000105158.2) carrying the upstream EcoRI-NotI and downstream BglII sequence was inserted into the pCAG-EGFP plasmid (Kaname and Huxley, 2001) cleaved by EcoRI and BglII. The resulting pCAG-Mos plasmid was then digested by EcoRI and NotI, and the mCherry sequence carrying the upstream EcoRI and downstream NotI sequence was inserted to create an in-frame mCherry-Mos fusion. The final pCAG-mCherry-Mos plasmid was verified by restriction digestion and sequencing. SalI and HindIII were used to release the transgene cassette. The 3636 bp construct was gel purified using the QIAGEN gel extraction kit according to the manufacturer's instructions, repurified using the QIAGEN PCR kit according to the manufacturer's instructions, and diluted to 10 ng/μl in embryo-certified water. Transgenic mice were generated in the transgenic facility of the Czech Phenogenomics Center of the Institute of Molecular Genetics by injecting linearized DNA into the male pronuclei of C57BL/6 1-cell embryos. Transgene-positive mice were identified by PCR (primer sequences are shown in Table EV1). Two founder animals were obtained, the lineage with better

detectable mCherry fluorescence was expanded and used for experiments in this work.

### Pkr (Eif2ak2) mutant mice

The Eif2ak2 (commonly known as Pkr) mutant model was produced in the Czech Centre for Phenogenomics at the Institute of Molecular Genetics ASCR using Cas9-mediated deletion of Eif2ak2 exons 2 (containing the first AUG) to 5 coding for dsRNA-binding domains (amino acids: 1–165). Sequences of guide RNAs were Pi1B: 5'-GTGTTTCCAACCCACCACAGG in the intron 1 and Pi5B: 5'-GGATCATTGTTGGTACACAGG in the intron 5 (yielding 7338-nt deletion). To produce guide RNAs, synthetic 128 nt guide DNA templates including T7 promoter, 18nt sgRNA and tracrRNA sequences were in vitro transcribed using the mMES-SAGE mMACHINE T7 Transcription Kit (Ambion) and purified using the mirPremier™ microRNA Isolation Kit (Sigma). The Cas9 mRNA was in vitro transcribed from pSpCas9 plasmid (PX165; Addgene plasmid #48137) using Ambion mMESSAGE mMA-CHINE T7 Transcription Kit, and purified using the RNeasy Mini kit (Qiagen). A sample for microinjection was prepared by mixing two guide RNAs in water (25 ng/μl for each) together with Cas9 mRNA (100 ng/μl). Five picoliters of the mixture were micro-injected into male pronuclei of C57Bl/6 zygotes and transferred into pseudo-pregnant recipient mice. PCR genotyping was performed on tail biopsies from 4-weeks-old animals. A positive founder which transmitted the mutant allele to F$_1$ was back-crossed with C57Bl/6NCrl animals for at least five generations before using in experiments.

Knock-out allele was detected using mPKR_i1_Fwd: 5'-GCCTTGTTTTGACCATAAATGCCG and mPKR_E6_Rev: 5'-GTGACAACGCTAGAGGATGTTCCG primers giving a 552 bp product (wild-type allele is too long to be amplified). Wild-type allele was detected using mPKR_i1_Fwd and mPKR_E2_gen_Rev: 5'-TGGCTACTCCGTGCATCTGG primers yielding a 404 bp product.

## Dicer$^{\Delta HEL1/wt}$ phenotype analysis

The phenotype analysis of animals with the ICR outbred background at the Czech Centre for Phenogenomics utilized the established phenotyping pipeline (https://mousephenotype.org/impress/index). The full report including description of the tests is provided in the Source Data for EV1.

## Plasmids for RNAi assay

Plasmids for RNAi assay were introduced in detail in our previous work (Demeter et al, 2019). Briefly, three dsRNA-expressing plasmids (MosIR, Lin28IR, and Elavl2IR) use inverted repeats, which efficiently induced RNAi in oocytes of transgenic mice (Flemr et al, 2014; Chalupnikova et al, 2014; Stein et al, 2003). In each transfection, one of these plasmids is combined with a non-targeted firefly luciferase reporter pGL4-SV40 (Promega; for simplicity referred to as FL) and a Renilla luciferase reporter carrying a Mos cognate sequence complementary to MosIR-expressed dsRNA. All non-commercial plasmids are available from Adgene with details about their construction and sequence. Plasmid sequences (annotated Genbank format) were provided previously (Demeter et al, 2019).

## Cell culture and transfection

Mouse ESCs were cultured in 2i-LIF media: KO-DMEM (Gibco) supplemented with 15% fetal calf serum (Sigma), 1x L-Glutamine (Thermo Fisher Scientific), 1x non-essential amino acids (Thermo Fisher Scientific), 50 µM β-Mercaptoethanol (Gibco), 1000 U/mL LIF (Isokine), 1 µM PD0325901, 3 µM CHIR99021 (Selleck Chemicals), penicillin (100 U/mL), and streptomycin (100 µg/mL). Mouse 3T3 cells were maintained in DMEM high glucose (Sigma) supplemented with 10% fetal calf serum, penicillin (100 U/mL), and streptomycin (100 µg/mL). For transfection, cells were plated on a 24-well plate, grown to 50% density, and transfected using Lipofectamine 3000 (Thermo Fisher Scientific) according to the manufacturer's protocol. For luciferase assay, cells were co-transfected with 100 ng of each FL and RL reporter plasmids, 400 ng of a dsRNA-expressing plasmid and, eventually, 400 ng of a plasmid expressing a tested factor per well. The total amount of transfected DNA was kept constant (1 µg/well) using pBluescript stuffer. Cells were collected for analysis 48 h post-transfection. For qPCR analysis, mESCs and 3T3 cells were transfected with 500 ng of dsRNA-expressing plasmid; the total amount of transfected DNA was kept constant (750 ng/well) using pBluescript stuffer. Cells were collected for analysis at 12, 24, 48, 72 h post-transfection.

## Luciferase assay

Dual luciferase activity was measured according to Hampf M. and Gossen M. (Hampf and Gossen, 2006) with some modifications. Briefly, cells were washed with PBS and lysed in PPTB lysis buffer (0.2% v/v Triton X-100 in 100 mM potassium phosphate buffer, pH 7.8). A 3–5 µl aliquots were used for measurement in 96-well plates using Modulus Microplate Multimode Reader (Turner Biosystems). First, firefly luciferase activity was measured by adding 50 µl substrate (20 mM Tricine, 1.07 mM $(MgCO_3)_4 \cdot Mg(OH)_2 \cdot 5H_2O$, 2.67 mM $MgSO_4$, 0.1 mM EDTA, 33.3 mM DTT, 0.27 mM Coenzyme A, 0.53 mM ATP, 0.47 mM D-Luciferin, pH 7.8) and signal was integrated for 10 s after 2 s delay. Signal was quenched by adding 50 µl Renilla substrate (25 mM $Na_4PP_i$, 10 mM Na-Acetate, 15 mM EDTA, 500 mM $Na_2SO_4$, 500 mM NaCl, 1.3 mM $NaN_3$, 4 µM Coelenterazine, pH 5.0) and *Renilla* luciferase activity was measured for 10 s after 2 s delay.

## Microscopy

Mice organs were harvested, rinsed with ice-cold PBS, and placed on ice. Organ images were acquired on a Zeiss Axio Zoom.V16 stereo microscope equipped with a Zeiss Axiocam 512 mono camera and analyzed with the Zeiss Zen 2.5 lite software. 1× 0.25 NA objective, the fluorescence filter sets GFP (excitation wavelength: 488/emission wavelength: 509) + RFP (excitation wavelength: 590/emission wavelength: 612) and the Bright Field optics were used for image acquisition. Background fluorescence levels for each organ were determined using WT littermates.

## qPCR

### mESCs and 3T3 cells

Mouse ESCs and 3T3 cells transfected with pCAG-EGFP-MosIR plasmid were collected at 12, 24, 48, 72 h post transfection. Cells were washed with PBS and total RNA was isolated by RNeasy Plus Mini Kit (Qiagen). RNA quality was verified by 1% Agarose gel electrophoresis. 800 ng and 1 µg of total RNA (respectively) was reverse transcribed using LunaScript RT SuperMix Kit (New England Bioloabs) according to the manufacturer's instructions and 1 µl cDNA was used as a template for a 10 µl qPCR reaction. qPCR was performed on LightCycler 480 (Roche) and the Maxima SYBR Green qPCR master mix (Thermo Fisher Scientific) was used for the qPCR reaction. qPCR was performed in technical triplicates for each biological sample. Average Ct values of the technical replicates were normalized to the housekeeping genes *Hprt*, *B2m*, and *Alas1* using the ΔΔCt method. A list of the primers used for qPCR is provided in Table EV1.

### Adult organs

For qPCR analysis, organs from adult mice (9–13 weeks old) were harvested, washed in PBS, and homogenized in Qiazol lysis reagent (Qiagen) and total RNA was isolated by phenol–chloroform extraction according to the manufacturer's protocol. 1 µg of total RNA was reverse transcribed using LunaScript RT SuperMix Kit (New England Biolabs) according to the manufacturer's instructions and 1 µl cDNA was used as a template for a 10 µl qPCR reaction. qPCR was performed on LightCycler 480 (Roche) and the Maxima SYBR Green qPCR master mix (Thermo Fisher Scientific) was used for the qPCR reaction. qPCR was performed in technical triplicates for each biological sample. Average Ct values of the technical replicates were normalized to the housekeeping genes *Hprt*, *B2m*, and *Alas1* using the ΔΔCt method. A list of the primers used for qPCR is provided in Table EV1.

## RNA sequencing

### ESC small RNA sequencing (RNA-seq)

Cells were plated on 6-well plates and grown to 80% density. Cells were transfected with 2 µg/well of pCAG-EGFP-MosIR plasmid and cultured for 48 h. Cells were washed with PBS, homogenized in Qiazol lysis reagent (Qiagen) and total RNA was isolated by Qiazol-chloroform extraction and ethanol precipitation method (Toni et al, 2018). RNA quality was verified by Agilent 2100 Bioanalyzer. Small RNA libraries were constructed using NEBNext Multiplex Small RNA Library Prep Set for Illumina (New England Biolabs) according to the manufacturer's protocol. Small RNA libraries were size selected on 6% PAGE gel, a band of 140–150 bp was cut from the gel and RNA was extracted using Monarch® Genomic DNA Purification Kit. Quality of the libraries was assessed by Agilent 2100 bioanalyzer. Libraries were sequenced on the Illumina HiSeq2000 platform at the Genomics Core Facility at EMBL.

### E15.5 small RNA-seq

E15.5 embryos were removed from the uterus and washed in PBS. The yolk sac was taken for genotyping and embryos were transferred into RNAlater (Thermo Fisher Scientific). Embryos were homogenized in Qiazol lysis reagent (Qiagen) and total RNA was isolated by Qiazol-chloroform extraction and ethanol precipitation method (Toni et al, 2018). Small RNA libraries were constructed using Nextflex Small RNA-seq kit v3 for Illumina (Perkin Elmer) according to the manufacturer's protocol; 3′ adapter ligation was performed overnight at 20 °C, 15 cycles were used for PCR amplification and NextFlex beads were used for size selection. Final libraries were sequenced by 75-nucleotide single-end reading using the Illumina NextSeq500/550 platform at the core genomics facility of IMG.

### Adult organ small RNA-seq

For small-RNA-seq analysis, organs from adult mice (9–13 weeks old) were harvested, washed in PBS, and homogenized in Qiazol lysis reagent (Qiagen) and total RNA was isolated by phenol–chloroform extraction according to the manufacturer's protocol. Small-RNA libraries were prepared using the NextFlex Small-RNA-seq v3 kit (Amplicon) according to the manufacturer's protocol; 3′ adapter ligation was performed overnight at 20 °C, 15–18 cycles were used for PCR amplification and gel purification was performed for size selection. For gel purification, libraries were separated on a 2.5% agarose gel using 1× lithium borate buffer and visualized with ethidium bromide. The 140–160 bp fraction was cut off the gel and DNA was isolated using the MinElute Gel Extraction Kit (Qiagen). Final libraries were sequenced by 75-nucleotide single-end reading using the Illumina NextSeq500/550 platform at the core genomics facility of IMG.

The list of small RNA-seq libraries from adult brain, heart, liver, muscle, spleen, and thymus is in Table EV2. Next to age-matched wild-type controls (C57Bl/6NCrl background), we sequenced four distinct genotypes on the C57Bl/6NCrl background, which included a transgene Tg(CAG-EGFP-MosIR) and one of the four genotypes: $Dicer^{SOM/wt}$ $Pkr^{+/-}$, $Dicer^{SOM/wt}$ $Pkr^{-/-}$, $Dicer^{\Delta HEL1/wt}$ $Pkr^{+/-}$, $Dicer^{\Delta HEL1/wt}$ $Pkr^{-/-}$. $Dicer^{SOM/wt}$ animals provided another control for phenotype analyses because the $Dicer^{SOM}$ allele has the same sequence as the $Dicer^{\Delta HEL1}$ allele except of the HEL1 sequence.

### Bioinformatic analyses

RNA-seq data (Table EV2) were deposited in the Gene Expression Omnibus database under accession numbers GSE243016 (reviewer access token: svgpsgamxvcjrad) and GSE242871 (reviewer access token: gjuhyqeglfsnvgh).

### Mapping of small RNA-seq data

Small RNA-seq reads were trimmed in two rounds using fastx-toolkit version 0.0.14 (http://hannonlab.cshl.edu/fastx_toolkit) and cutadapt version 1.8.3 (Martin, 2011). First, 4 random bases were trimmed from left side:

```
fastx_trimmer -f 5 -i {INP}.fastq -o {TMP}.fastq
```

Next, NEXTflex adapters were trimmed. In addition, the N-nucleotides on ends of reads were trimmed and reads containing more than 10% of the N-nucleotides were discarded:

```
cutadapt --format = "fastq" --front =
"GTTCAGAGTTCTACAGTCCGACGATCNNNN" --adapter =
"NNNNTGGAATTCTCGGGTGCCAAGG" --error-rate=0.075
--times=2 --overlap=14 --minimum-length=12 --max-
n = 0.1 --output = "${TRIMMED}.fastq" --trim-n
--match-read-wildcards ${TMP}.fastq
```

Trimmed reads were mapped to the mouse (mm10) genome with following parameters:

```
STAR --readFilesIn ${TRIMMED}.fastq.gz --runThreadN 4
--genomeDir ${GENOME_INDEX} --genomeLoad
LoadAndRemove --readFilesCommand unpigz -c
--readStrand Unstranded --limitBAMsortRAM
```

```
20000000000 --outFileNamePrefix ${FILENAME}
--outReadsUnmapped Fastx --outSAMtype BAM
SortedByCoordinate --outFilterMultimapNmax 99999
--outFilterMismatchNoverLmax 0.1
--outFilterMatchNminOverLread 0.66
--alignSJoverhangMin 999 --alignSJDBoverhangMin 999
```

### miRNA expression analyses

Mapped reads were counted using program featureCounts (Liao et al, 2014). Only reads with lengths 19–25 nt were selected from the small RNA-seq data:

```
featureCounts -a ${ANNOTATION_FILE} -F ${FILE}
-minOverlap 15 -fracOverlap 0.00 -s 1 -M -O -fraction -T
8 ${FILE}.bam
```

The GENCODE gene set (Frankish et al, 2019) was used for the annotation of long RNA-seq data. For small RNA-seq data analysis, the miRBase 22.1 (Kozomara et al, 2019) miRNA annotation was combined with published mirtron annotation (Ladewig et al, 2012). Small RNA expression was analyzed as described previously (Zapletal et al, 2022). Statistical significance and fold changes in gene expression were computed in R using the DESeq2 package (Love et al, 2014). Genes were considered to be significantly up- or down-regulated if their corresponding p-adjusted values were smaller than 0.05. The DESeq2 baseMean and fold changes were plotted and visualized by home-made R scripts as described previously (Zapletal et al, 2022).

## Data availability

RNA sequencing data were deposited to Gene Expression Omnibus (GEO) with the following accession numbers GSE243016 (https://www.ncbi.nlm.nih.gov/geo/query/acc.cgi?acc=GSE243016) and GSE242871 (https://www.ncbi.nlm.nih.gov/geo/query/acc.cgi?acc=GSE242871). The full-list of samples is provided in the Table EV2. Original codes were deposited at https://github.com/fhorvat/2023.RNAi_activation.

The source data of this paper are collected in the following database record: biostudies:S-SCDT-10_1038-S44319-024-00148-z.

## Peer review information

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

## Acknowledgements

We thank Vladimir Benes and EMBL sequencing facility for help with RNA-seq experiments, and Kristian Vlahovicek for providing hardware support for bioinformatics analyses. Development of *Dicer^{SOM}* and *Dicer^{ΔHEL1}* mouse models was funded from the European Research Council under the European Union's Horizon 2020 research and innovation program (grant agreement No. 647403, D-FENS). Main funding was provided by the Czech Science Foundation EXPRO grant 20-03950X. Additional support was provided by the Ministry of Education, Youth, and Sports (MEYS) project NPU1 LO1419. Financial support of VB, FH, and MK was in part provided by the Charles University in a form of a PhD student fellowship; this work will be in part used to fulfil requirements for a PhD degree and hence can be considered "school work". The authors used services of the Czech Centre for Phenogenomics at the Institute of Molecular Genetics supported by the Czech Academy of Sciences RVO 68378050 and by the project LM2018126 and LM2023036 Czech Centre for Phenogenomics provided by Ministry of Education, Youth and Sports (MEYS)of the Czech Republic. We also acknowledge services of the Light Microscopy Core Facility, IMG, Prague, Czech Republic, supported by MEYS – LM2023050 and RVO – 68378050-KAV-NPUI. Computational resources were provided by the e-INFRA CZ project (ID:90254), supported by the Ministry of Education, Youth and Sports of the Czech Republic and by the ELIXIR-CZ project (ID:90255), part of the international ELIXIR infrastructure.

## Author contributions

**Valeria Buccheri**: Data curation; Formal analysis; Investigation; Visualization; Writing—review and editing. **Josef Pasulka**: Data curation; Software; Formal analysis; Investigation; Visualization; Writing—review and editing. **Radek Malik**: Conceptualization; Data curation; Formal analysis; Supervision; Validation; Investigation; Project administration; Writing—review and editing. **Zuzana Loubalova**: Investigation; Writing—review and editing. **Eliska Taborska**: Data curation; Investigation; Writing—review and editing. **Filip Horvat**: Data curation; Software; Investigation; Writing—review and editing. **Marcos Iuri Roos Kulmann**: Investigation; Writing—review and editing. **Irena Jenickova**: Investigation; Writing—review and editing. **Jan Prochazka**: Resources; Data curation; Formal analysis; Supervision; Investigation; Visualization; Project administration; Writing—review and editing. **Radislav Sedlacek**: Supervision; Funding acquisition; Writing—review and editing. **Petr Svoboda**: Conceptualization; Resources; Data curation; Formal analysis; Funding acquisition; Validation; Writing—original draft; Writing—review and editing.

Source data underlying figure panels in this paper may have individual authorship assigned. Where available, figure panel/source data authorship is listed in the following database record: biostudies:S-SCDT-10_1038-S44319-024-00148-z.

## Disclosure and competing interests statement

The authors declare no competing interests.

# Expanded View Figures

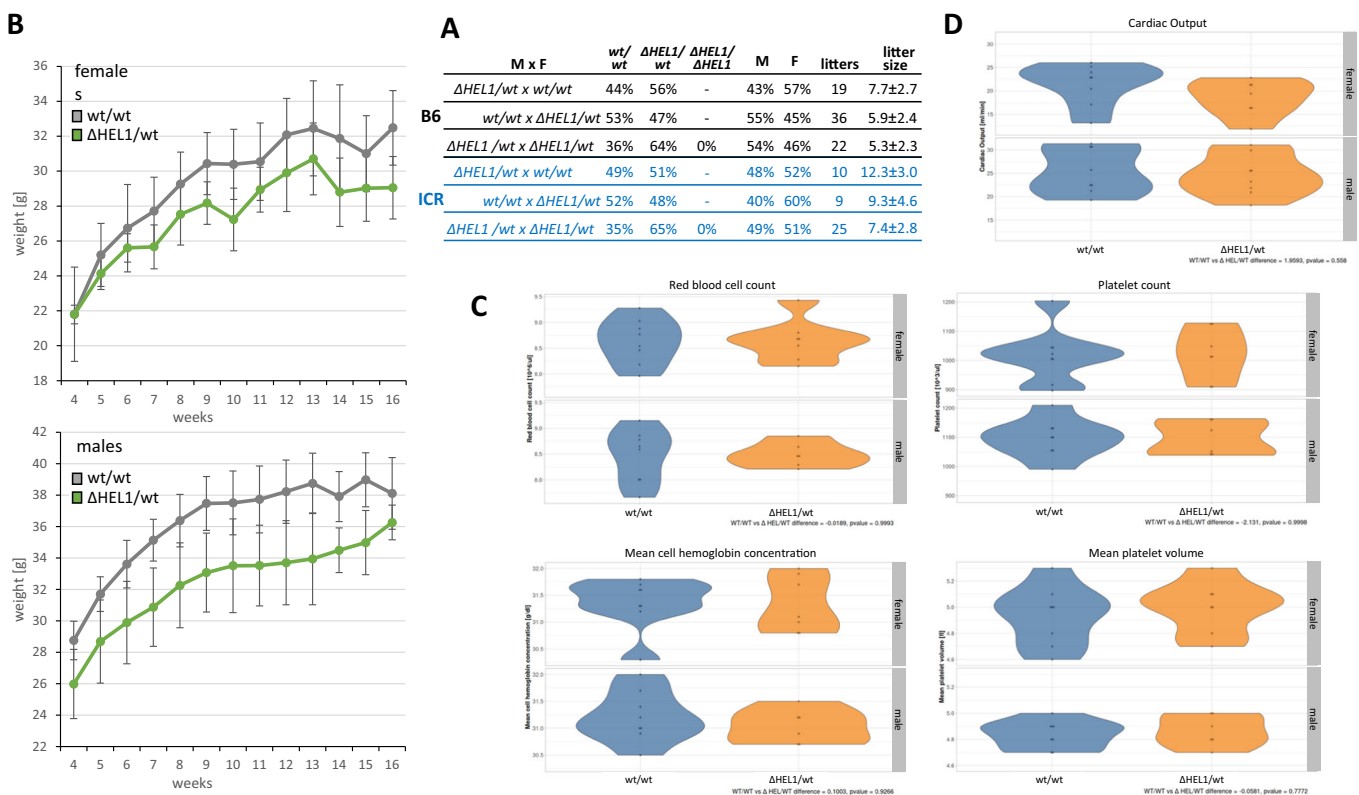

**Figure EV1. Selected phenotype features.**

(A) Breeding performance of heterozygous mutants expressed as percentages. (B) Growth curves of males and females on the ICR background. 8 animals were analyzed for each genotype. Data are presented as mean +/− SD error bars. (C) Dicer$^{\Delta HEL1/wt}$ animals have normal cardiac output. (D) Blood parameters disrupted in Dicer$^{\Delta HEL1/\Delta HEL1}$ (Zapletal et al, 2022) are normal in Dicer$^{\Delta HEL1/wt}$ mice. For details on the methodology, see the full phenotyping report in the Source Data for EV1. Source data are available online for this figure.

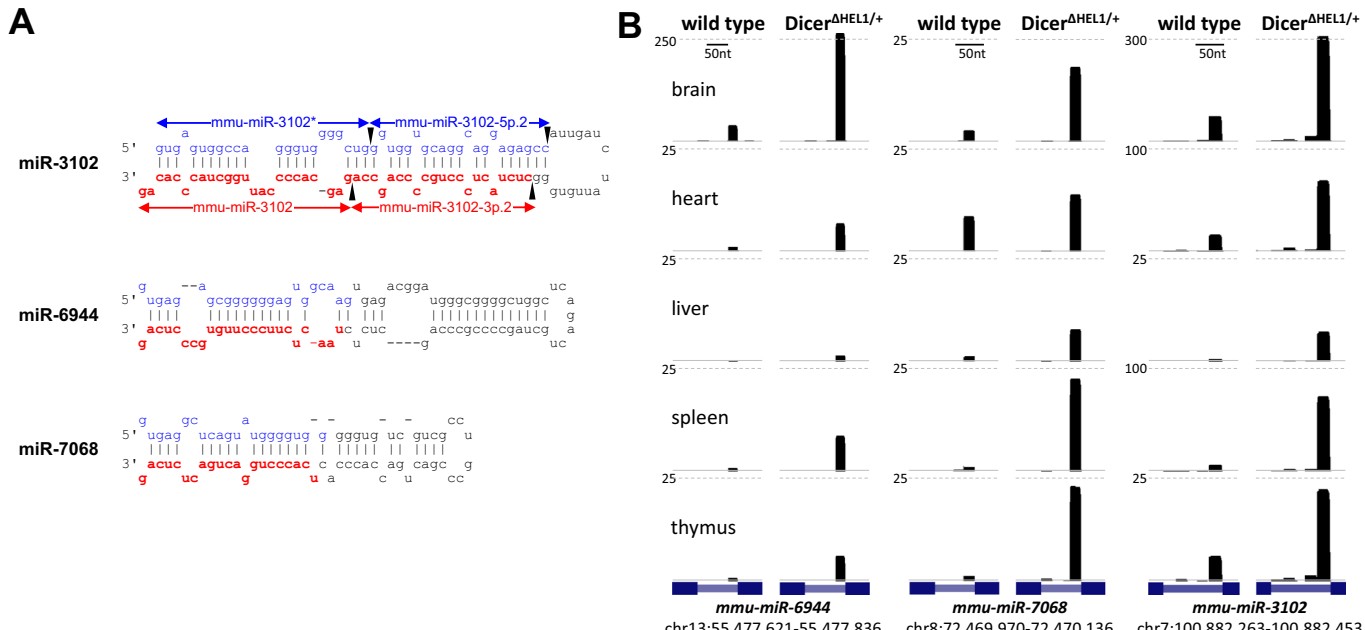

**Figure EV2. Supplementary data for miRNome changes.**

(A) Most upregulated mirtrons. Mirtron precursor schemes were adapted from structures presented in miRBase (Kozomara et al, 2019). (B) UCSC browser snapshots showing abundance of 21–23 nt reads in mirtron loci in normal and ΔHEL1 mice. The vertical scale is in counts per million (CPM) of 18–32 nt reads.

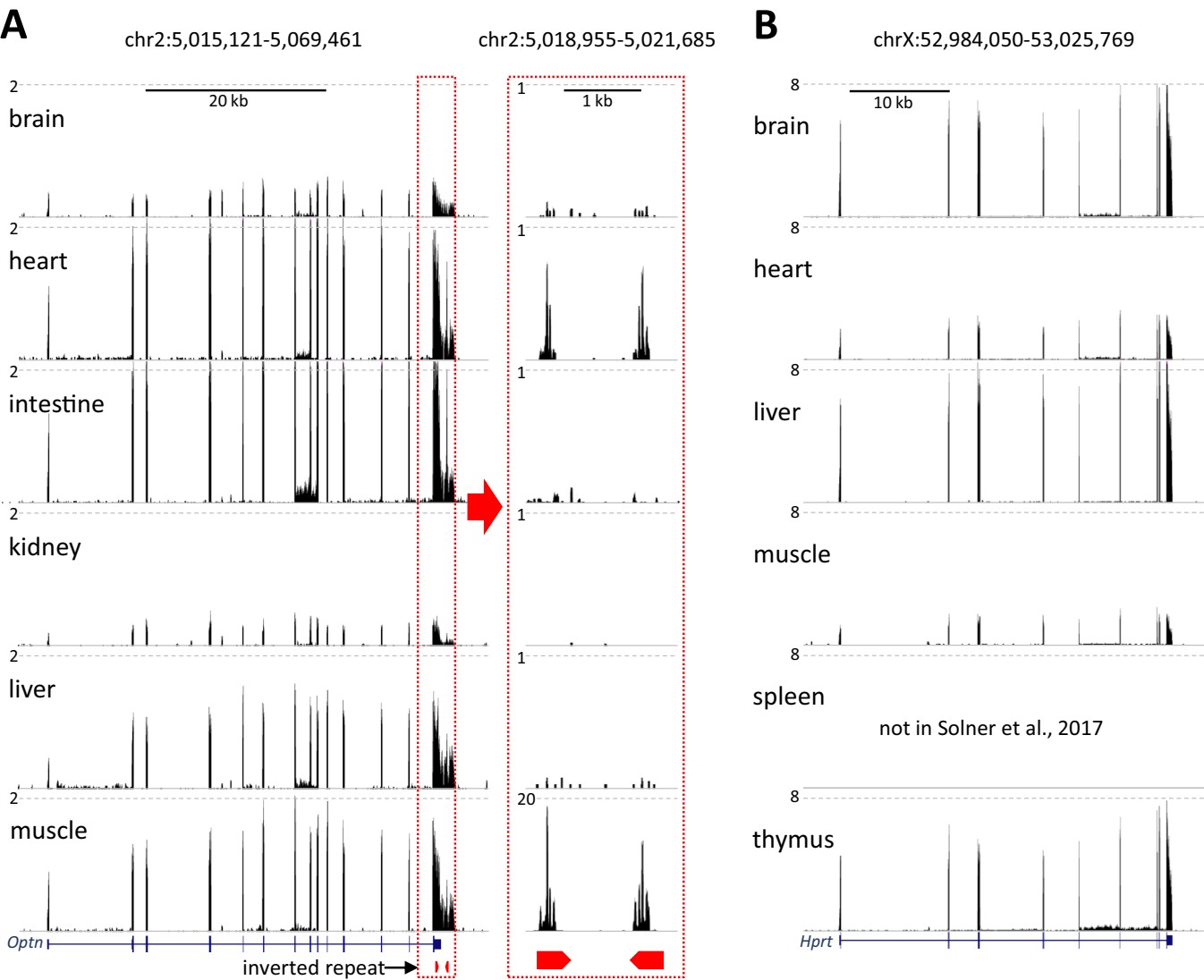

**Figure EV3. Supplementary data for siRNA analysis.**

(A) *Optn* expression and siRNA production in different organs in wild-type mice. On the left side is a UCSC browser snapshot of *Optn* transcript expression in different organs using publicly available RNA-seq libraries from different organs (Sollner et al, 2017). The vertical scale is counts per million of reads (CPM). Next to it are 21–23 nt RNAs from small RNA sequencing data from the same organs (Isakova et al, 2020) mapped into the *Optn* inverted repeat region (red pentagons). The vertical scale is in counts per million of 19–32 nt reads. (B) UCSC browser snapshot of *Hprt* expression in different organs in wild-type mice using publicly available RNA-seq libraries from different organs (Sollner et al, 2017).

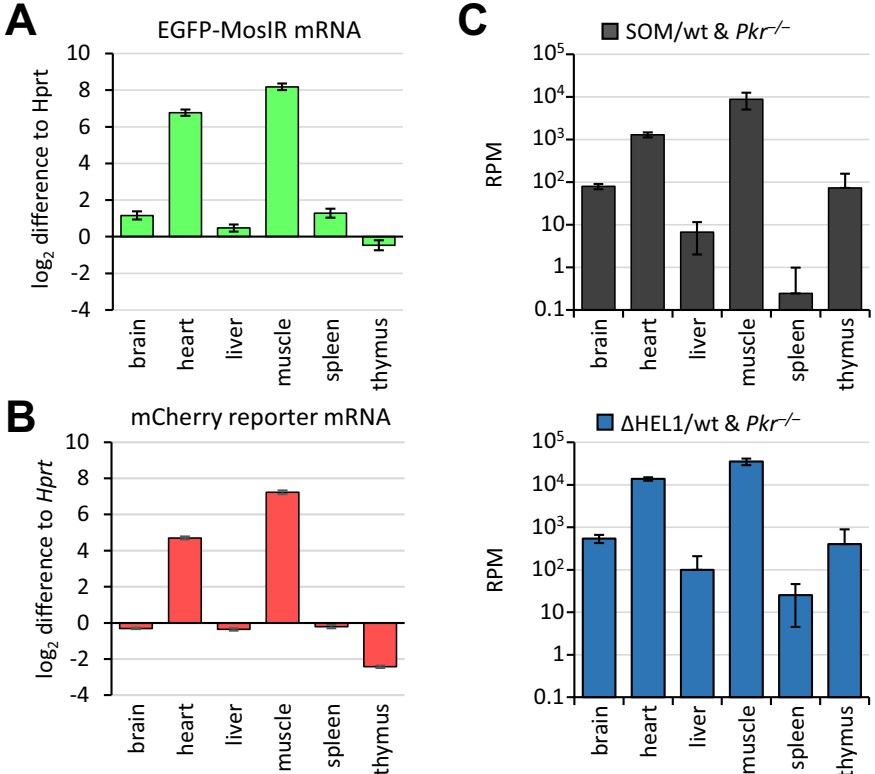

**Figure EV4. Supplementary data for RNAi analysis—logscale presentation of transcript and siRNA abundance.**

(A) qPCR analysis of CAG-EGFP-MosIR transgene expression in *Dicer*$^{SOM/wt}$ *Pkr*$^{-/-}$ organs from Fig. 5C presented as log$_2$ relative difference to *Hprt* expression. Data come from three (muscle and spleen) or five (other organs) biological replicates analyzed in technical triplicates. (B) qPCR analysis of CAG-mCherry-Mos transgene expression in *Dicer*$^{SOM/wt}$ *Pkr*$^{-/-}$ organs of a single animal from Fig. 6B presented as log$_2$ relative difference to *Hprt* expression. qPCR analysis was done in technical triplicates. (C) MosIR endo-siRNA abundance—shown is 21–23 nt RNA abundance in reads per million (RPM) of all 18–32 nt mapped small RNA reads. All bar graphs depict mean $+/-$ SD error bars. Source data are available online for this figure.

