## [Peer Review File · EMBO Reports]

Functional canonical RNAi in mice expressing a truncated Dicer isoform and long dsRNA

Valeria Buccheri, Josef Pasulka, Radek Malik, Zuzana Loubalova, Eliska Taborska, Filip Horvat, Marcos Iuri Roos Kulmann, Irena Jenickova, Jan Prochazka, Radislav Sedlacek, and Petr Svoboda

Corresponding author(s): Petr Svoboda (svobodap@img.cas.cz)

Review Timeline:

Transfer Date:	10th Jan 24
Editorial Decision:	16th Jan 24
Revision Received:	10th Mar 24
Editorial Decision:	2nd Apr 24
Revision Received:	9th Apr 24
Editor's Email:	11th Apr 24
Author's Response to the Email:	15th Apr 24
Accepted:	15th Apr 24

Editor: Esther Schnapp

Transaction Report: This manuscript was transferred to EMBO reports following peer review at The EMBO Journal.

Referee #1:

The manuscript by Buccheri et al. describes how RNAi can be activated in vivo in mice by making them heterozygous for a specific Dicer allele. The allele named HEL, does not support miRNA function well, and in homozygous state was already shown to be detrimental for development (due to miRNA defects). Heterozygous animals, however, are surprisingly healthy and display only minor miRNA defects (which are limited to defects in mirtron processing). This allows for examination of RNAi effectiveness, both endogenous as well as exogenous (induced by non-natural trigger dsRNA).

The bottom line is that siRNA production is boosted in these animals, and that RNAi mediated gene repression can be achieved. At the same time, the authors conclude that endogenous siRNA levels remain below levels needed for activity, whereas transgene-induced RNAi can reach such levels. In the latter case, repression is still not very strong (roughly 50%) and may well be affected by numerous other factors, besides siRNA abundance.

Finally, the authors look at the impact of Pkr, a kinase that is triggered by dsRNA. They see hardly any effect of loss of Pkr in their mice, leading them to conclude that siRNA levels are first and foremost limited by levels of dsRNA.

This is important work because it is still debated to what extent RNAi may be relevant in mice (and in general in mammals). Antiviral effects have been described, but these have not seen any follow-up that would support those studies. The current manuscript provides a nicely systematic study on the potential effectiveness of RNAi in mice. Even if the conclusions are somewhat sobering (RNAi being not very potent) they are important for two main reasons:

- this is a unique study that benchmarks RNAi activity in vivo in mice
- it establishes a system in which potential natural functions of RNAi can be much better studied in mice (and in parallel to a functioning miRNA system)

The conclusions are overall well supported by the data and I support publication in EMBO J. I would request some adjustments for clarity, as some parts of the manuscript were hard to follow. Notably, this applies to the section where the authors discuss PKR for the first time (experiments in ESCs, page 6). The results were presented in a rather convoluted way. For instance, it is first mentioned that 21-23 nucleotide small RNAs are increased, and much later also those that are outside this size range. The logic was not presented very well here. Ideally, being a manuscript that presents a system to use RNAi in mice in specific tissues, an application would be shown, however, to ask for that in mice is unreasonable. Can't the authors use primary cells to show that RNAi against a number of genes will work much better in these than in wild-type cells? Just to broaden the tested sequences and thereby enhance the impact of the work.

Referee #2:

In this manuscript, the authors demonstrate that Dicer[deltaHEL1/+] mice have minimal impact on the apparent phenotype and the miRNA pathway, but functional RNAi can be detected in heart when triggering dsRNAs are over-expressed. Overall, the experiments are well conducted, and the results are clear. However, the biological implication of the current observations remains somewhat ambiguous.

Specific points:

1. The heterogeneous condition of Dicer[deltaHEL1/+] is not naturally occurring but artificially created. I'm not sure if the observed RNAi in this system, which require both a copy of Dicer[deltaHEL1] allele and over-expression of dsRNAs, will really serve as "an unprecedented platform for addressing claims about its biological roles." Of course, it is technically possible to perform, e.g., viral infection and other experiments in Dicer[deltaHEL1/+] mice, but how should we interpret the results? (e.g., enhanced anti-viral activity in Dicer[deltaHEL1/+] does not formally affirm or deny the biological role in wild-type animals.) At least, the current text does not sufficiently clarify how this platform can be effectively used to explore the true biological roles of RNAi in mammals.
2. Could the authors discuss why heart, but not other organs, can produce abundant siRNAs and exhibit detectable RNAi?
3. Could the minor growth reduction observed in Dicer[deltaHEL1/+] be attributed to the upregulation of mitrons? How are the expression levels of their predicted target genes affected?
4. Fig. 1F appears to be missing.
5. Figure numbers should be clearly shown in the respective figure layouts, at least at the peer-review stage.

Referee #3:

This work by Buccheri et al. assesses the effect of boosting endogenous dsRNA dicing in vivo. Previous work from the team demonstrated that oocytes encode a specific Dicer isoform, DicerO, better at generating endo-siRNAs via dsRNA cleavage. Mice engineered to only express DicerO, but not canonical Dicer (DicerHel1/Hel1) are embryonic lethal. In this work, the authors study the production of siRNAs from dsRNA in viable DicerWT/Hel1 mice. They show that WT/Hel1 mice indeed produce more endo-siRNAs from endogenous dsRNA as well as from a transgenic locus overexpressing dsRNA (MosIR). MosIR siRNAs are most abundant in the heart, in which they silence a Mos-mCherry construct. In this artificial context (two transgenes with strong promoters), dsRNA-driven silencing only occurs with abundant siRNAs. I list thereafter comments regarding the soundness of the experimental approaches. The authors use a fairly artificial system to monitor dsRNA-driven RNAi, which may explain the results. I am not sure

what these results tell us regarding endo-siRNAs, their generation and silencing efficiency. As the authors mention, a key question that could be addressed with the WT/He1 strain is the potential antiviral effect of increasing dsRNA degradation. It would make sense to enrich the present study with such results.

Major points

- 1) In Fig 1F, the authors claim that the expression of dsRNA-coding plasmid is affected by PKR. This should be verified in the system by monitoring plasmid transcription by RT-qPCR.
- 2) As mentioned by the authors, the effect of PKR on siRNA production from dsRNA needs to be performed by comparing small RNAs in a WT/SOM;PKR+/- background to WT/He1; PKR+/- . Fig 2A should be replaced by Fig S2A not to risk confusing the reader and the text should be edited accordingly.
- 3) Fig 4E, this data alone does not document the role of ADAR in editing dsRNA. What are the levels of ADAR in each organ? Without assessing editing in an ADAR null background as a control, the authors should tone down their claim.

Minor points

- 1) Fig 1B, present as percentages instead of raw numbers
- 2) Fig 2D would gain to be expressed as fold change over control, to be able to quantify the changes.

Referee #1:

The manuscript by Buccheri et al. describes how RNAi can be activated in vivo in mice by making them heterozygous for a specific Dicer allele. The allele named HEL, does not support miRNA function well, and in homozygous state was already shown to be detrimental for development (due to miRNA defects). **Heterozygous animals, however, are surprisingly healthy and display only minor miRNA defects (which are limited to defects in mirtron processing). This allows for examination of RNAi effectiveness, both endogenous as well as exogenous (induced by non-natural trigger dsRNA).**

The bottom line is that siRNA production is boosted in these animals, and that RNAi mediated gene repression can be achieved. At the same time, the authors conclude that endogenous siRNA levels remain below levels needed for activity, whereas transgene-induced RNAi can reach such levels. In the latter case, repression is still not very strong (roughly 50%) and may well be affected by numerous other factors, besides siRNA abundance.

Finally, the authors look at the impact of Pkr, a kinase that is triggered by dsRNA. They see hardly any effect of loss of Pkr in their mice, leading them to conclude that siRNA levels are first and foremost limited by levels of dsRNA.

This is important work because it is still debated to what extent RNAi may be relevant in mice (and in general in mammals). Antiviral effects have been described, but these have not seen any follow-up that would support those studies. The current manuscript provides a nicely systematic study on the potential effectiveness of RNAi in mice. Even if the conclusions are somewhat sobering (RNAi being not very potent) they are important for two main reasons:

- this is a unique study that benchmarks RNAi activity in vivo in mice
- it establishes a system in which potential natural functions of RNAi can be much better studied in mice (and in parallel to a functioning miRNA system)

The conclusions are overall well supported by the data and I support publication in EMBO J. I would request some adjustments for clarity, as some parts of the manuscript were hard to follow.

This issue can be addressed by rewriting/clarifying the text in the revised version.

Notably, this applies to the section where the authors discuss PKR for the first time (experiments in ESCs, page 6). The results were presented in a rather convoluted way. For instance, it is first mentioned that 21-23 nucleotide small RNAs are increased, and much later also those that are outside this size range. The logic was not presented very well here.

This issue can be addressed by rewriting/clarifying the text in the revised version.

Ideally, being a manuscript that presents a system to use RNAi in mice in specific tissues, an application would be shown, however, to ask for that in mice is unreasonable.

We have tested antiviral effects *[REDACTED: Author's response with unpublished data.]* but there were no reproducible antiviral effects with any of the viruses. *[REDACTED: Author's response with unpublished data.]* However, the work on viruses is a part of another manuscript and we prefer to present virus-related data together in a single separate publication. First, in this manuscript, we want to provide separate detailed characterization of dHEL1 heterozygotes and their ability to produce siRNAs and activate RNAi. Second, viruses represent a different type of substrates when compared to endogenous dsRNA, which in mammalian cells usually has inaccessible termini (i.e. with at least one long overhang).

Third, long dsRNA-induced RNAi and antiviral RNAi are distinct because the virus is replicating and Dicer cleavage can by itself directly suppress (i.e. in AGO2-independent manner) the viral infection. This contrasts with monitoring reporter silencing in trans with MosIR-derived siRNAs.

Can't the authors use primary cells to show that RNAi against a number of genes will work much better in these than in wild-type cells? Just to broaden the tested sequences and thereby enhance the impact of the work.

It is not clear what primary cells are meant – dHEL1/primary cells from mice? The primary problem is that for testing RNAi this way, a long inverted repeat expression vector would have to be produced for every targeted gene and the experimental design would require to deal with transfection efficiency of the primary cells or making stable lines from primary cells. Of note is that we have previously tested embryonic stem cells homozygous for dHEL1 using long inverted repeats and other dsRNA-expressing systems against ELAVL2 and LIN28A/B (10.26508/Isa.201800289). Silencing of endogenous genes was not apparent upon transient transfection when expression is measured by qPCR on the entire population but it became visible when most transfected cells were FACS-sorted and reanalyzed. We could perform another set of experiments on dHEL/+ ESCs but we do not see how that would strengthen the manuscript as it is even more artificial than what has been criticized by reviewers 2 and 3 and would only confirm that high dsRNA expression generates enough siRNA for silencing. However, if this experiment would be required for the revision, we should be able to perform it in the second half of January.

Referee #2:

In this manuscript, the authors demonstrate that Dicer[deltaHEL1/+] mice have minimal impact on the apparent phenotype and the miRNA pathway, but functional RNAi can be detected in heart when triggering dsRNAs are over-expressed. **Overall, the experiments are well conducted, and the results are clear. However, the biological implication of the current observations remains somewhat ambiguous.**

We politely disagree. There are several biological implications of our results when considering that RNAi effects in dHEL1/+ mice were analyzed next to wild type mice (and other mutants).

- unlike in oocytes, expression of dHEL1 variant in murine somatic cells does not bring high abundance of endo-siRNAs in somatic cells. There is negligible amount of long dsRNA, which could be converted into siRNAs.
- in the presence of the full-length Dicer, miRNA biogenesis fidelity is largely preserved, that is a major observation with implications for developing a truncated Dicer variant into a therapeutic tool.
- canonical RNAi induced by endogenous dsRNA expression can be achieved *in vivo*
- genetically modified animals have clearly higher Dicer activity everywhere, are phenotypically thoroughly characterized, and are available for anyone willing to test their antiviral properties. Off note is that both, SOM and dHEL1 alleles lack the ability to produce the proposed AviD Dicer variant (10.1126/science.abg2264) and thus provide a perfect control for *in vivo* analysis of its loss.

Specific points:

1. The heterogeneous condition of Dicer[deltaHEL1/+] is not naturally occurring but artificially created.

Naturally, genetic engineering of Dicer allele creates a genetically altered experimental system – notably the system is a viable mouse with minor alteration of the miRNA pathway, which is consistent with the structural alteration of Dicer as we have described previously (10.1016/j.molcel.2022.10.010). To be fair, one should also consider that transiently transfected HeLa cells or ESCs are even more artificial systems than a mouse model, which directly shows that modified Dicer activity has a minimal effect on its physiology. dHEL1/+ is thus as little artificial in terms of affecting the whole living organism as possible and is the least artificial *in vivo* model next to the wild type mouse.

Furthermore, this heterogeneous condition to some extent exists in mouse, rat, and hamster oocytes as evidenced by published RNA-seq and qPCR data (10.1101/gr.216150.116, see Fig. 5 and the figure on the right). In oocytes, each Dicer allele produces simultaneously both, the full-length and dHEL1 transcripts in a species-specific ratio.

In addition, when the truncated Dicer variant becomes expressed at the beginning of oocyte development, combined levels of both Dicer variants are also present there.

I'm not sure if the observed RNAi in this system, which require both a copy of Dicer[deltaHEL1] allele and over-expression of dsRNAs, will really serve as "an unprecedented platform for addressing claims about its biological roles."

We are convinced that this platform:

1) is unprecedented – nobody moved from artificial cell cultures models to the organismal level in mammals like we did. We provide the same set up for RNAi experiments like artificial cell culture

experiments. The primary difference is that our system is constrained by physiological levels of Dicer expression and abundance of stably expressed long dsRNA. Unlike artificially created cell culture experiments, our system is genetic one and put into a mammal. Thus, the fact that most organs do not exhibit RNAi provides valuable information about limiting conditions for RNAi *in vivo*. Furthermore, one should distinguish the dHEL1/+ mouse with increased siRNA production capacity from the dsRNA-expressing transgene and RNAi sensor transgene. All mice are freely available for anyone willing to study endogenous or antiviral RNAi *in vivo*.

2) allows to see which claims about endogenous RNAi obtained from cell culture experiments are reproducible in *in vivo*. This includes the function of the proposed AviD Dicer variant (10.1126/science.abg2264) as both, SOM and dHEL1 alleles cannot produce it. dHEL1/+ mouse has ~10x higher capacity to process long dsRNA, which did not seem to be saturated by one of the strongest available transgenic promoters. This is thus a great model, which should show significantly improved RNAi if used for testing of previous claims concerning RNAi and dsRNA toxicity under different scenarios.

Of course, it is technically possible to perform, e.g., viral infection and other experiments in Dicer[deltaHEL1/+] mice, but how should we interpret the results? (e.g., enhanced anti-viral activity in Dicer[deltaHEL1/+] does not formally affirm or deny the biological role in wild-type animals.) At least, the current text does not sufficiently clarify how this platform can be effectively used to explore the true biological roles of RNAi in mammals.

Our results should be interpreted such that mammals have negligible RNAi activity and if siRNA-biogenesis is plugged into the miRNA pathway, it is efficient only with very high siRNAs levels. This also suggests that if functional RNAi would exist in some specific somatic cell-type, it would likely require additional adaptation(s) such as stabilization/availability of dsRNA substrate, uncoupling of AGO2 from miRNA-mediated silencing, suppressing target-mediated decay of AGO2-loaded small RNA etc. Notably, there is no enhanced anti-viral activity in dHEL1/+ animals.

We have tested antiviral effects [REDACTED: Author's response with unpublished data.] but there were no reproducible antiviral effects with any of the viruses. [REDACTED: Author's response with unpublished data.] However, the work on viruses is a part of another manuscript and we prefer to present virus-related data together in a single separate publication. First, in the current manuscript, we want to provide separate detailed characterization of dHEL1 heterozygotes and their

ability to produce siRNAs and activate RNAi. Second, viruses represent a different type of substrates when compared to endogenous dsRNA, which in mammalian cells usually has inaccessible termini (i.e. with at least one long overhang).

Third, long dsRNA-induced RNAi and antiviral RNAi are distinct because the virus is replicating and Dicer cleavage can by itself directly suppress (i.e. in AGO2-independent manner) the viral infection. This contrasts with monitoring reporter silencing in trans with MosIR-derived siRNAs.

2. Could the authors discuss why heart, but not other organs, can produce abundant siRNAs and exhibit detectable RNAi?

In the MosIR system, siRNA abundance primarily correlates with dsRNA transgene expression, which is high in the heart. Notably, we also see abundant siRNAs levels in the muscle, which was analyzed recently but not for all genotypes and thus not included into the original manuscript version. The MosIR expression is also very high there (we can provide/included these data when

required/requested). In additions, several additional factors could contribute to efficiency of RNAi: folding/unfolding of dsRNA, RNA editing, half-life (of AGO-loaded small RNA). These factors can be discussed in the revised version.

3. Could the minor growth reduction observed in *Dicer*[Δ HEL1/+] be attributed to the upregulation of mirtrons? How are the expression levels of their predicted target genes affected?

It could but it cannot be addressed in a reasonable time for a revision. While we have dysregulated transcriptomes of dHEL1/dHEL1 ESCs, in which we could analyze expression changes of predicted targets of these two mirtrons, this analysis cannot yield any conclusive results. This question can only be addressed experimentally by removing the two most upregulated mirtrons from the genome and then testing if that would rescue the minor growth defect in dHEL1/+ animals. However, we do not plan to do it, it is not worth of spending money, time and motivation of a PhD student on that.

4. Fig. 1F appears to be missing.

The panel F is in the middle of Fig. 1.

5. Figure numbers should be clearly shown in the respective figure layouts, at least at the peer-review stage.

This issue can be addressed in the revised version easily.

Referee #3:

This work by Buccheri et al. assesses the effect of boosting endogenous dsRNA dicing *in vivo*. Previous work from the team demonstrated that oocytes encode a specific Dicer isoform, DicerO, better at generating endo-siRNAs via dsRNA cleavage. Mice engineered to only express DicerO, but not canonical Dicer (DicerHel1/Hel1) are embryonic lethal. In this work, the authors study the production of siRNAs from dsRNA in viable DicerWT/Hel1 mice. They show that WT/Hel1 mice indeed produce more endo-siRNAs from endogenous dsRNA as well as from a transgenic locus overexpressing dsRNA (MosIR). MosIR siRNAs are most abundant in the heart, in which they silence a Mos-mCherry construct. In this artificial context (two transgenes with strong promoters), dsRNA-driven silencing only occurs with abundant siRNAs. I list thereafter comments regarding the soundness of the experimental approaches.

The authors use a fairly artificial system to monitor dsRNA-driven RNAi, which may explain the results.

We would like to point out that the system stands on careful and proven design of its elements. It is not very clear what the reviewer means when saying that the system monitoring RNAi explains the results. We demonstrate that dHEL1 increases by an order of magnitude siRNAs production from long dsRNA and that with the highest expression of MosIR in heart comes high abundance of siRNAs and efficient silencing. That lower MosIR expression does not generate enough siRNAs to silence a reporter expressed at a level comparable with *Hprt* is not an artifact but demonstration of limits of RNAi *in vivo*.

I am not sure what these results tell us regarding endo-siRNAs, their generation and silencing efficiency.

These results provide valuable benchmark parameters for assessment of functionality of endogenous canonical RNAi. They demonstrate that endo-siRNAs in the level of hundreds of CPMs cannot efficiently silence a reporter expressed at a level comparable to *Hprt* mRNA. These results show what it takes to achieve 10^4 RPM siRNA abundance *in vivo* (which likely corresponds to nanomolar cytoplasmic siRNA concentration when compared with Let-7 abundance - $10.1093/\text{nar/gkaa543}$ and Table 1). This is actually matching siRNA concentrations used in knock-down experiments. For the miRNA field this finding would not be surprising (especially when considering TDMD) and for RNAi field it should serve as a reminder that mammalian RNAi is to a large extent a feature of the miRNA pathway. We believe these are important outcomes.

As the authors mention, a key question that could be addressed with the WT/Hel1 strain is the potential antiviral effect of increasing dsRNA degradation. It would make sense to enrich the present study with such results.

We have tested antiviral effects [REDACTED: Author's response with unpublished data.] but there were no reproducible antiviral effects with any of the viruses. [REDACTED: Author's response with unpublished data.] However, the work on viruses is a part of another manuscript and we prefer to present virus-related data together in a single separate publication. First, we want to provide separate detailed characterization of dHEL1 heterozygotes and their ability to produce

siRNA and activate RNAi. Second, viruses represent a different type of substrates when compared to endogenous dsRNA, which in mammalian cells usually has inaccessible termini (i.e. with at least one long overhang).

Third, long dsRNA-induced RNAi and antiviral RNAi are distinct because the virus is replicating and Dicer cleavage can directly suppress (i.e. in AGO2-independent manner) the viral infection. This contrasts with monitoring reporter silencing in trans with MosIR-derived siRNAs.

Major points

1) In Fig 1F, the authors claim that the expression of dsRNA-coding plasmid is affected by PKR. This should be verified in the system by monitoring plasmid transcription by RT-qPCR.

We have RNA-seq data from the endpoint of the experiment suggesting increased levels of MosIR transcript, which could be added to the work. Importantly, we have published previously that PKR affects expression of transiently-transfected plasmids: 10.1371/journal.pone.0043283 and 10.1371/journal.pone.0087517. The effect primarily takes place at the level of translation but that might change stability and availability of MosIR transcript for Dicer processing. This is thus clearly a complex issue concerning an artificial cell culture scenario. We would revise the text, add quantification of data from RNA-seq. We could also perform qPCR analysis of MosIR expression in the absence of PKR but it is hard to envision how any result would affect the outcome of the work.

2) As mentioned by the authors, the effect of PKR on siRNA production from dsRNA needs to be performed by comparing small RNAs in a WT/SOM;PKR+/- background to WT/Hel1; PKR+/- . Fig 2A should be replaced by Fig S2A not to risk confusing the reader and the text should be edited accordingly.

We did most of the experiments on PKR-/- background as we expected that we might see more efficient RNAi there. We thus prefer to show the genotype with the maximum expected RNAi in the main figure. However, we can switch the figures as suggested or add additional comparisons of the effect of PKR (loss) on siRNA production.

3) Fig 4E, this data alone does not document the role of ADAR in editing dsRNA. What are the levels of ADAR in each organ? Without assessing editing in an ADAR null background as a control, the authors should tone down their claim.

Analysis shown in Fig. 4E documents variable MosIR dsRNA editing, which can be supported by additional data. Fig. 4E shows that A/G change is the most common sequence change observed in sequenced small RNA (relative to all other nucleotide changes combined) and that it varies among organs. As the analysis looks at editing in the inverted repeat sequence, it provides good support that the observed A->G change is indeed coming from dsRNA editing by ADARs. We can provide additional supporting evidence from this experiment, which is consistent with our previous analyses where we showed that RNA fragments originating from ssRNA regions (EGFP) do not have this feature (10.1371/journal.pone.0087517). We can provide RNA expression levels for ADARs from the literature, ENCODE, BioGPs, or the Protein Atlas, but these levels would not validate or invalidate A/G and other nucleotide changes seen in RNA-seq data from predicted single-stranded and double-stranded RNA. Editing of MosIR in *Adar* null background cannot be realistically assessed as it would require *Adar1* & 2 null mutants plus at least three other additional homozygous genetic modifications to make *Adar* mutant mice viable plus adding dHEL1 allele and the MosIR transgene to the genotype.

Minor points

1) Fig 1B, present as percentages instead of raw numbers

We will add %.

2) Fig 2D would gain to be expressed as fold change over control, to be able to quantify the changes.

There is no Figure 2D. If the reviewer means 2B, we propose to add numerical data into the supplement as a table.

Dear Petr,

Thank you for the transfer of your manuscript and point-by-point response to EMBO reports. I have now discussed all with my colleagues here, and we would like to invite you to address the referee concerns as you suggest in your point-by-point response.

Please do add additional comparisons of the effect of PKR loss on siRNA production, as you propose. Please also modify the title, as it is somewhat misleading. Regarding the viral data, these are clearly very relevant, and we think it would be best if your other ms with the viral data was posted on BioRxiv so that it can be discussed and the work cited in your ms submitted here. What should not happen is, that your other ms is published first, as I understand that it uses the same dHEL1/+ mice. Please let me know at what stage the other ms is and what you plan to do.

I would thus like to invite you to revise your manuscript with the understanding that the referee concerns must be fully addressed and their suggestions taken on board. Please address all referee concerns in a complete point-by-point response. Acceptance of the manuscript will depend on a positive outcome of a second round of review. It is EMBO reports policy to allow a single round of major revision only and acceptance or rejection of the manuscript will therefore depend on the completeness of your responses included in the next, final version of the manuscript.

We realize that it is difficult to revise to a specific deadline. In the interest of protecting the conceptual advance provided by the work, and given that no major experimentation is required, we recommend a revision within 2 months (17th March 2024). Please discuss the revision progress ahead of this time with the editor if you require more time to complete the revisions.

- 1) A data availability section providing access to data deposited in public databases is missing. If you have not deposited any data, please add a sentence to the data availability section that explains that.
- 2) Your manuscript contains statistics and error bars based on $n=2$. Please use scatter blots in these cases. No statistics should be calculated if $n=2$.

5) a complete author checklist, which you can download from our author guidelines

<<https://www.embopress.org/page/journal/14693178/authorguide>>. Please insert information in the checklist that is also reflected in the manuscript. The completed author checklist will also be part of the RPF.

6) Please note that all corresponding authors are required to supply an ORCID ID for their name upon submission of a revised manuscript (<<https://orcid.org/>>). Please find instructions on how to link your ORCID ID to your account in our manuscript tracking system in our Author guidelines <<https://www.embopress.org/page/journal/14693178/authorguide#authorshipguidelines>>

7) Before submitting your revision, primary datasets produced in this study need to be deposited in an appropriate public database (see <https://www.embopress.org/page/journal/14693178/authorguide#datadeposition>). Please remember to provide a reviewer password if the datasets are not yet public. The accession numbers and database should be listed in a formal "Data Availability" section placed after Materials & Method (see also <https://www.embopress.org/page/journal/14693178/authorguide#datadeposition>). Please note that the Data Availability Section is restricted to new primary data that are part of this study. * Note - All links should resolve to a page where the data can be accessed. *
If your study has not produced novel datasets, please mention this fact in the Data Availability Section.

I look forward to seeing a revised form of your manuscript when it is ready.

Best wishes,
Esther

Referee #1:

The conclusions are overall well supported by the data and I support publication in EMBO J. I would request some adjustments for clarity, as some parts of the manuscript were hard to follow.

Notably, this applies to the section where the authors discuss PKR for the first time (experiments in ESCs, page 6). The results were presented in a rather convoluted way. For instance, it is first mentioned that 21-23 nucleotide small RNAs are increased, and much later also those that are outside this size range. The logic was not presented very well here.

To address this comment, we thoroughly revised the manuscript and reorganized figures and figure panels to provide a simpler and more direct narrative. Briefly, we first introduce the phenotype of dHEL1/+ animals, then we report effects on miRNAs, then we summarize changes in siRNA production (endogenous first, MosIR-derived next), and then we bring up analysis of RNAi knock-down effects. PKR is now discussed in the siRNA production part where we compare siRNA levels in transient transfection experiments with the genetic model in mice – we added new data suggesting that in transient transfection, PKR loss results in higher plasmid expression - that could be the main reason for previous observations and provides a simple explanation why that effect is not observed in mice *in vivo*.

Ideally, being a manuscript that presents a system to use RNAi in mice in specific tissues, an application would be shown, however, to ask for that in mice is unreasonable.

We have tested antiviral effects with five different viruses but there were no reproducible antiviral effects with any of them. However, the work on viruses is a part of another manuscript and we prefer to present virus-related data together in a single separate publication. First, in this manuscript, we want to provide separate detailed characterization of dHEL1 heterozygotes and their ability to produce siRNAs and activate canonical RNAi. Second, viruses represent a different type of substrates when compared to endogenous dsRNA, which in mammalian cells usually has inaccessible termini (i.e. with at least one long overhang). Third, long dsRNA-induced RNAi and antiviral RNAi are distinct because the virus is replicating and Dicer cleavage can by itself directly suppress (i.e. in AGO2-independent manner) the viral infection. This contrasts with monitoring reporter silencing in *trans* with MosIR-derived siRNAs.

Can't the authors use primary cells to show that RNAi against a number of genes will work much better in these than in wild-type cells? Just to broaden the tested sequences and thereby enhance the impact of the work.

The primary problem with testing RNAi on dHEL1/+ primary cells from mice as suggested is that a long inverted repeat expression vector would have to be produced for every targeted gene and the experimental design would require either making stable lines from primary cells or dealing with transfection efficiency of the primary cells and PKR effects. It is not clear how this would increase impact of the work as knockdown can generally be achieved with a different inverted repeat in mammalian cells - we have previously tested embryonic stem cells homozygous for dHEL1 using long inverted repeats and other dsRNA-expressing systems against ELAVL2 and LIN28A/B (10.26508/Isa.201800289). Silencing of endogenous genes was not apparent upon transient transfection when expression was measured by qPCR on the entire population but it became visible when most transfected cells were FACS-sorted and reanalyzed. To increase the number of instances of RNAi, the revised version includes analysis of muscle, the second organ where RNAi was detected and was accompanied with high siRNA abundance.

Referee #2:

In this manuscript, the authors demonstrate that Dicer[deltaHEL1/+] mice have minimal impact on the apparent phenotype and the miRNA pathway, but functional RNAi can be detected in heart when triggering dsRNAs are over-expressed. Overall, the experiments are well conducted, and the results are clear. However, the biological implication of the current observations remains somewhat ambiguous.

We politely disagree that biological implications remain ambiguous. There are several biological implications, which should not be overlooked.

- In the presence of the full-length Dicer, miRNA biogenesis fidelity is largely preserved when siRNA biogenesis is boosted by a truncated Dicer variant. This is a major observation with implications for developing a truncated Dicer variant into a therapeutic tool, e.g. in macular degeneration (doi.org/10.1073/pnas.1909761117).
- dHEL1/+ mice, which have demonstrably higher Dicer activity in all examined organs, were phenotypically thoroughly characterized and show that mice can tolerate activation of RNAi analogous to that in mouse oocytes even when present ubiquitously and constitutively. That model is now available to anyone for testing *in vivo* their ambiguous claims about biological roles of mammalian RNAi.
- We show that PKR does not have an effect on endogenous canonical RNAi even when dsRNA is overexpressed. This is a significant observation, especially when combined with showing that increased siRNA abundance in transiently transfected Pkr^{-/-} cells rather comes from increased expression of dsRNA-expressing plasmid than further increased efficiency of Dicer cleavage.
- We show that natural endo-siRNAs remain abysmally low-abundant even in the presence of the dHEL1 Dicer variant, which shows that there is no excess of substrates, which could be processed into highly abundant siRNAs upon increasing Dicer activity. Mirtron precursors appeared to be the only available substrates, which could give rise to medium-abundant small RNAs upon increasing Dicer activity.
- Our work is the first that provides some insight into siRNA abundance necessary to induce efficient RNAi response from long dsRNA *in vivo* and it shows that it is at par with highly abundant miRNAs. Our work thus shows that efficient targeting cellular transcripts *in vivo* may require rather high threshold for siRNA abundance, as indicated by siRNA levels in heart and muscle, where we observe knockdown.

Specific points:

1. The heterogeneous condition of Dicer[deltaHEL1/+] is not naturally occurring but artificially created.

While the genetic engineering of a Dicer allele creates a genetically altered experimental system for testing effects of more or less equimolar level of two Dicer protein variants, expression of two Dicer protein variants at different ratios is naturally occurring in rodent oocytes where each Dicer allele produces two Dicer variants in a species-specific ratio. Consequently, there are different expression levels of the two Dicer variants in mouse, rat, and hamster oocytes as evidenced by published RNA-seq and qPCR data (10.1101/gr.216150.116, see Fig. 5 and the figure below). Also, stoichiometric amount of truncated and full-length Dicer variants must also occur at some point at the beginning of oocyte development when the truncated Dicer variant becomes expressed and activates RNAi.

Moreover, it has been proposed that mammalian stem cells produce AviD, a spliced Dicer variant with altered helicase domain (10.1126/science.abg2264). If true, this would represent another example of co-expression of two Dicer variants.

Thus, we are convinced that this reviewer's point does not invalidate our work. In fact, we are convinced that among possible experimental designs to boost Dicer

activity *in vivo*, our solution is as close as possible to mimicking RNAi activation, which could occur by evolution of somatic expression of dHEL1 variant or by a simple deletion fusing in frame exons 2 and 7. Importantly, our Dicer engineering relies on mimicking an existing Dicer variant with proven biological role and solved structure (10.1016/j.molcel.2022.10.010). The resulting dHEL1/+ model is viable and fertile, which makes it the least artificial *in vivo* mouse model with enhanced RNAi.

I'm not sure if the observed RNAi in this system, which require both a copy of Dicer[deltaHEL1] allele and over-expression of dsRNAs, will really serve as "an unprecedented platform for addressing claims about its biological roles."

In the revised version the statement was reworded to: "*Dicer*^{ΔHEL1/wt} mice with enhanced canonical RNAi offer a unique platform for examining potential of mammalian RNAi *in vivo* and its limits." We believe that this accurately reflects the value of the collection of genetically modified animals from this paper. We made the original claim because the set of mouse modifications is unprecedented indeed. There is no other *in vivo* mouse model providing three genetically separated components:

- enhanced siRNA biogenesis (dHEL/+)
- long dsRNA expression (MosIR)
- targeted reporter (mCherry-Mos)

This offers to set up an RNAi experiment in mice *in vivo* like it has been done in cell culture experiments. The obvious difference is that our system is constrained by physiological levels of Dicer expression and abundance of stably expressed long dsRNA because this is a genetic system in a living mammal, unlike artificial cell culture experiments relying on transfection. Thus, the fact that most organs do not exhibit RNAi, still provides valuable information about limiting conditions for RNAi *in vivo*.

Regarding "addressing claims about biological roles", we want to reiterate that we provide here:

- For testing antiviral RNAi claims, we provide a viable dHEL/+ mouse with enhanced Dicer activity and siRNA biogenesis boosted an order of magnitude above that in the wild type. The last decade brought many claims about mammalian antiviral RNAi but lacked animal RNAi models.
- For testing the role of AviD Dicer variant, we provide a viable mouse model expressing full-length Dicer but lacking AviD isoform (SOM/SOM mouse) for which SOM/+ and dHEL/+ mice would be excellent controls.
- For testing claims concerning toxic dsRNA accumulation in pathologies, dHEL/+ mouse is an excellent model for resistance against toxic dsRNA accumulation while MosIR provides a dsRNA-expressing transgene .

All mice are freely available and we sincerely wish our colleagues would take our mouse models and would demonstrate their claims about biological roles of RNAi on these mice *in vivo*.

Of course, it is technically possible to perform, e.g., viral infection and other experiments in Dicer[deltaHEL1/+] mice, but how should we interpret the results? (e.g., enhanced anti-viral activity in Dicer[deltaHEL1/+] does not formally affirm or deny the biological role in wild-type animals.) At least, the current text does not sufficiently clarify how this platform can be effectively used to explore the true biological roles of RNAi in mammals.

Our results should be interpreted such that mammals have negligible canonical RNAi activity and if siRNA-biogenesis is plugged into the miRNA pathway, it is efficient only with very high siRNAs levels. This also suggests that if functional RNAi would exist in some specific somatic cell-type, it would likely require additional adaptation(s) such as stabilization/availability of dsRNA substrate, uncoupling of AGO2 from miRNA-mediated silencing, suppressing target-mediated decay of AGO2-loaded small RNA etc. Notably, there was no enhanced anti-viral activity in dHEL1/+ animals tested on five different viruses. We observed presence of virus-derived endosRNA and achieved antiviral effects in cultured ESCs with increased expression of dHEL1 Dicer. But not in animals. The work on viruses is a part of another manuscript and we prefer to present virus-related data together in a

single separate publication. First, in the current manuscript, we want to provide separate detailed characterization of dHEL1 heterozygotes and their ability to produce siRNAs and activate canonical RNAi. Second, viruses represent a different type of substrates when compared to endogenous dsRNA, which in mammalian cells usually has inaccessible termini (i.e. with at least one long overhang). Third, long dsRNA-induced RNAi and antiviral RNAi are distinct because the virus is replicating and Dicer cleavage can by itself directly suppress (i.e. in AGO2-independent manner) the viral infection. This contrasts with monitoring reporter silencing in trans with MosIR-derived siRNAs.

2. Could the authors discuss why heart, but not other organs, can produce abundant siRNAs and exhibit detectable RNAi?

In the MosIR system, siRNA abundance primarily correlates with dsRNA transgene expression, which is high in the heart. Notably, we see even more abundant siRNAs levels and even stronger RNAi in the muscle, which was analyzed recently. The MosIR expression in muscle is the highest and this correlates with the highest Mos siRNA abundance we have ever observed. Also, with this MosIR expression, full-length Dicer apparently generates enough siRNAs to observe an RNAi effect in SOM/+ animal. We now include muscle data in the revised manuscript. Thus, since RNAi correlates with CAG expression, we think that the key factor is very high MosIR dsRNA expression. We cannot rule out additional contributing factors, some of which are discussed in the discussion (e.g. RISC composition).

3. Could the minor growth reduction observed in Dicer[deltaHEL1/+] be attributed to the upregulation of mirtrons? How are the expression levels of their predicted target genes affected?

It is possible but addressing this question is beyond the scope of this manuscript. For analysis of mirtron targets, we have available only transcriptomes of dHEL1/dHEL1 ESCs. These cells have considerably perturbed transcriptomes but also miRNomes because of their dHEL1/dHEL1 genotype and thus are not suitable for determining how mirtron upregulation affects gene expression. Reviewer's question would best be addressed experimentally by removing the two most upregulated mirtrons from the genome and then testing if that would rescue the minor growth defect in dHEL1/+ animals. However, we do not plan to do it, the lab does not have resources for that.

4. Fig. 1F appears to be missing.

The panel F was in the middle of Fig. 1. However, figures were reorganized for the revised version and Fig.1 one does not have this panel anymore, siRNA biogenesis in dHEL1 mutants was moved to Fig. 3.

5. Figure numbers should be clearly shown in the respective figure layouts, at least at the peer-review stage.

Figure numbers were added into figure layouts

Referee #3:

This work by Buccheri et al. assesses the effect of boosting endogenous dsRNA dicing *in vivo*. Previous work from the team demonstrated that oocytes encode a specific Dicer isoform, DicerO, better at generating endo-siRNAs via dsRNA cleavage. Mice engineered to only express DicerO, but not canonical Dicer (DicerHel1/He1) are embryonic lethal. In this work, the authors study the production of siRNAs from dsRNA in viable DicerWT/He1 mice. They show that WT/He1 mice indeed produce more endo-siRNAs from endogenous dsRNA as well as from a transgenic locus overexpressing dsRNA (MosIR). MosIR siRNAs are most abundant in the heart, in which they silence a Mos-mCherry construct. In this artificial context (two transgenes with strong promoters), dsRNA-driven silencing only occurs with abundant siRNAs. I list thereafter comments regarding the soundness of the experimental approaches.

The authors use a fairly artificial system to monitor dsRNA-driven RNAi, which may explain the results.

It is apparent that RNAi is observed only with high siRNA abundance, which is apparently restricted to heart (and muscle) because of dsRNA overexpression from the CAG promoter but presence/absence of RNAi and levels of MosIR, siRNA, and mCherry-Mos target still provide valuable information about RNAi *in vivo*. That lower MosIR expression does not generate enough siRNAs to silence a reporter expressed at a level comparable with *Hprt* is not an artifact but demonstration of limits of RNAi *in vivo*. We would like to point out that setting up canonical RNAi *in vivo* with transgenes would be artificial in any case. Importantly, the system stands on careful and proven design of its elements where MosIR architecture follows that of *Optn*, a natural siRNA substrate. The same ubiquitous promoter for dsRNA and target assures expression in all tissues and a similar ratio of expression of both elements. qPCR analysis shows that dsRNA trigger is expressed more than the targeted reporter.

I am not sure what these results tell us regarding endo-siRNAs, their generation and silencing efficiency.

These results provide valuable benchmark parameters for assessment of functionality of endogenous canonical RNAi. They demonstrate that:

- Natural endo-siRNAs have negligible levels and are hardly biologically relevant.
- A single dHEL1 allele boosts siRNA biogenesis by approximately an order of magnitude but natural endo-siRNAs still have negligible levels.
- A single dHEL1 allele processing transgene-expressed dsRNA has capacity to make siRNAs in the range of four orders of magnitude.
- siRNA biogenesis from expressed long dsRNA *in vivo* is not affected by presence/absence of PKR
- With muscle data added, we now show that RNAi is observed with endo-siRNAs in higher thousands of RPMs and above. While this siRNA abundance coming from an overexpressed transgene *in vivo*, it is in line with synthetic siRNA concentrations in RNAi experiments in cell culture and abundance of functionally important miRNAs. We bring up this point in discussion.

Our findings should not be surprising in the miRNA field (especially when considering TDMD) and for RNAi field it should serve as a reminder that mammalian RNAi is to a large extent a feature of the miRNA pathway. We believe these are important outcomes telling a simple and sobering narrative.

As the authors mention, a key question that could be addressed with the WT/He1 strain is the potential antiviral effect of increasing dsRNA degradation. It would make sense to enrich the present study with such results.

We have tested antiviral effects with five different viruses but there were no reproducible antiviral effects with any of them. We could observe presence of virus-derived endo-siRNA (small RNA-seq & TraPR column) and achieve antiviral effects in cultured ESCs with increased expression of dHEL1 Dicer. But not in animals. The work on viruses is a part of another manuscript and we prefer to present virus-related data together in a single separate publication. First, we want to provide

separate detailed characterization of dHEL1 heterozygotes and their ability to produce siRNA and activate RNAi. Second, viruses represent a different type of substrates when compared to endogenous dsRNA, which in mammalian cells usually has inaccessible termini (i.e. with at least one long overhang). Third, long dsRNA-induced RNAi and antiviral RNAi are distinct because the virus is replicating and Dicer cleavage can directly suppress (i.e. in AGO2-independent manner) the viral infection. This contrasts with monitoring reporter silencing in trans with MosIR-derived siRNAs.

Major points

1) In Fig 1F, the authors claim that the expression of dsRNA-coding plasmid is affected by PKR. This should be verified in the system by monitoring plasmid transcription by RT-qPCR.

In the revised manuscript, we provide the requested data in Fig. 4F, which show that loss of PKR results in 3-4-times increase of MosIR expression in ESCs and 3T3 cells.

2) As mentioned by the authors, the effect of PKR on siRNA production from dsRNA needs to be performed by comparing small RNAs in a WT/SOM;PKR+/- background to WT/Hel1; PKR+/- . Fig 2A should be replaced by Fig S2A not to risk confusing the reader and the text should be edited accordingly.

In the revised version of the manuscript, we now show in Fig. 2A comparison of *Dicer*^{ΔHEL1/wt} *Pkr*^{+/-} Tg(CAG-EGFP-MosIR) animals with small RNA profiles of normal age-matched C57Bl/6NCrl, which we have available for five organs and the comparison of *Dicer*^{ΔHEL1/wt} *Pkr*^{+/-} Tg(CAG-EGFP-MosIR) with *Dicer*^{SOM/wt} *Pkr*^{+/-} Tg(CAG-EGFP-MosIR), which shows reduced variability and upregulation of mirtrons in the panel 2B. The loss of PKR is then analyzed separately on Optn endo-siRNA (Fig. 3G) and Mos endo-siRNA (Fig. 4E). We believe that the revised version presents data in an easier-to-follow way.

3) Fig 4E, this data alone does not document the role of ADAR in editing dsRNA. What are the levels of ADAR in each organ? Without assessing editing in an ADAR null background as a control, the authors should tone down their claim.

We removed editing data from the manuscript because addition of muscle data and quantified overexpression of MosIR made interpretation of results ambiguous. While we are convinced that analysis of A->G change is a good proxy for detecting RNA editing by ADARs, minimal A->G changes in MosIR RNA fragments in heart and muscle could reflect saturation of editing mechanism with highly overexpressed long dsRNA (Fig. 4C) or low editing in these organs. However, literature search did not provide any coherent explanation of observed data and we thus decided that their ambiguous interpretation would not provide added value to the manuscript.

Editing of MosIR in *Adar* null background cannot be realistically assessed in near future because it would require *Adar1* & 2 null mutants plus at least three other additional homozygous genetic modifications to make *Adar* mutant mice viable plus adding dHEL1 allele and the MosIR transgene to the genotype.

Minor points

1) Fig 1B, present as percentages instead of raw numbers

As percentages would hide the scale of analysis, we provide the table with percentages in the supplement as Fig. S1A and original mouse numbers in the main figure 1B.

2) Fig 2D would gain to be expressed as fold change over control, to be able to quantify the changes.

There was no Figure 2D. If the reviewer meant 2B, a graph with fold changes instead of the heatmap would be too busy than the current heatmap. We could provide a table with fold change values if this would be considered important for readers.

Dear Petr,

Thank you for the submission of your revised manuscript to EMBO reports. We have now received the enclosed reports from the referees and I am happy to say that all support its publication now.

Only a few more minor editorial requests will need to be addressed before we can proceed with the official acceptance of your manuscript:

- Your ms has 5 main figures but the results and discussion sections are not combined. Please either add one more main figure, or combine the results and discussion sections (this is preferred) to publish your study as a short report.
- Please reduce the number of keywords to 5.
- The conflict of interest subheading needs to be corrected to "Disclosure and Competing Interests Statement"
- The author credits need to be removed from the ms file. All credits need to be entered during online ms submission.
- The Supplementary figures and tables need to be renamed to EV Figures (Figure EV1, etc.) and EV tables; and all callouts in the ms text need to be updated accordingly. All EV figures and tables need to be uploaded as individual files. Your Table S1 can either be an EV table, or a regular Table 1 in the methods section.
- Please upload all SD of one figure as one (if necessary zipped) folder, so that all SD are uploaded as one folder per figure.
- Methods needs to be corrected to Materials and Methods
- The manuscript sections should be in the following order: Title page - Abstract & Keywords - Introduction - Results - Discussion - Materials & Methods - Data Availability - Acknowledgments - Disclosure Statement & Competing Interests - References - Figure Legends - Tables with legends - Expanded View Figure Legends.
- I am not sure what to do with the "Supplementary File S1 A complete Dicer Δ HEL1/wt phenotyping report (html file)". Is this may be source data? Do you want to link this to the main ms file? May be this could be done as source data. Please let me know what you would like to do with this.
- Please provide the specific URLs for GSE242871 and GSE243016 datasets in the DAS (Data Availability Section).
- Please address these comments by our data editors:
 1. Please indicate the statistical test used for data analysis in the legend of figure 2a.
 2. Please note that information related to n is missing in the legends of figures 3a-b, g; 4e, supplementary figures 1b; 4a, c.
 3. Although 'n' is provided, please describe the nature of entity for 'n' in the legends of figures 3d-e; 5f.
 4. Please note that the error bar is not defined in the legend of supplementary figure 1b.
 5. Please note that the measure of center for the error bars needs to be defined in the legends of figures 1c; 3a-b, d-e, g; 4c, e-f; 5b, e-f, supplementary figures 4a-c.
 6. Please note that scale bar and its definition are missing for figures 5c-d.

I would like to suggest some minor changes to the last sentences of the abstract that needs to be written in present tense:

dsRNA expression from a transgene yields sufficient siRNA levels to induce RNAi in heart and muscle. Dicer Δ HEL1/wt mice with enhanced canonical RNAi offer a platform for examining potentials and limits of mammalian RNAi in vivo.

EMBO press papers are accompanied online by A) a short (1-2 sentences) summary of the findings and their significance, B) 2-3 bullet points highlighting key results and C) a synopsis image that is exactly 550 pixels wide and 200-600 pixels high (the height is variable). You can either show a model or key data in the synopsis image. Please note that text needs to be readable at the final size. Please send us this information along with the final manuscript.

Best wishes,
Esther

Referee #1:

The authors have revised the manuscript adequately, providing a more balanced and accurate descriptions and discussions. I support its publication.

Referee #2:

The authors significantly improved the manuscript and provided satisfactory answers to my comments. The conclusions drawn from the experiments are adequately phrased. This work contains interesting data for the field, I therefore recommend it for publication.

Referee #3:

I have gone through the manuscript and its revisions since the EMBO J review, and I believe all issues have been addressed sufficiently. I support publication in EMBO Rep as is.

Only a few more minor editorial requests will need to be addressed before we can proceed with the official acceptance of your manuscript:

- Your ms has 5 main figures but the results and discussion sections are not combined. Please either add one more main figure, or combine the results and discussion sections (this is preferred) to publish your study as a short report.

We split Fig. 3 to two figures, the paper has six figures now. Please, note that Fig. 5C (now 6C) had mirrorflipped heart, which was also corrected.

- Please reduce the number of keywords to 5.

OK

- The conflict of interest subheading needs to be corrected to "Disclosure and Competing Interests Statement"

OK

- The author credits need to be removed from the ms file. All credits need to be entered during online ms submission.

OK

- The Supplementary figures and tables need to be renamed to EV Figures (Figure EV1, etc.) and EV tables; and all callouts in the ms text need to be updated accordingly. All EV figures and tables need to be uploaded as individual files. Your Table S1 can either be an EV table, or a regular Table 1 in the methods section.

OK

- Please upload all SD of one figure as one (if necessary zipped) folder, so that all SD are uploaded as one folder per figure.

Reorganized into one archive with a folder per figure.

- Methods needs to be corrected to Materials and Methods

OK

- The manuscript sections should be in the following order: Title page - Abstract & Keywords - Introduction - Results - Discussion - Materials & Methods - Data Availability - Acknowledgments - Disclosure Statement & Competing Interests - References - Figure Legends - Tables with legends - Expanded View Figure Legends.

Reorganized accordingly

- I am not sure what to do with the "Supplementary File S1 A complete Dicer Δ HEL1/wt phenotyping report (html file)". Is this may be source data? Do you want to link this to the main ms file? May be this could be done as source data. Please let me know what you would like to do with this.

Ideally Source data for EV1

- Please provide the specific URLs for GSE242871 and GSE243016 datasets in the DAS (Data Availability Section).

- Please address these comments by our data editors:

1. Please indicate the statistical test used for data analysis in the legend of figure 2a.

This is RNA-seq analysis. Statistical significance comes from the algorithm used for analysis of differential expression - this is described in material and methods (Statistical significance and fold changes in gene expression were computed in R using the DESeq2 package).

2. Please note that information related to n is missing in the legends of figures 3a-b, g; 4e, supplementary figures 1b; 4a, c.

For data from RNA-sequencing, where sequencing samples are listed in the Table EV2 (all our samples are in triplicates), thus n is self-explanatory. This concerns figures 3a, d, e, f (formerly 3g), 5e (formerly 4e), and supplementary EV4C. Is thus really necessary to mention this in every panel? If you feel that yes, please, add it there, I feel this is becoming excessively redundant and disruptive for reading through figure legends. Regarding Fig. 3B, this is a compilation of several experiments. Information on n is in the figure legend. EV4A and B are just another display of data from Fig 5C and 6B where the main figure legend describes how the experiment was done explicitly (2 or 3 organs done in technical RT-qPCR triplicates).

3. Although 'n' is provided, please describe the nature of entity for 'n' in the legends of figures 3d-e; 5f.

This request is puzzling – for 3d-e n is obviously the number of RNA-seq replicates listed in the table EV1. Fig. 3 legend is already long (315 words). Is this really necessary?! Furthermore, 5f legend says already: White numbers indicate the number of organ pairs used for comparison. What else should we add?

4. Please note that the error bar is not defined in the legend of supplementary figure 1b.

Corrected

5. Please note that the measure of center for the error bars needs to be defined in the legends of figures 1c; 3a-b, d-e, g; 4c, e-f; 5b, e-f, supplementary figures 4a-c.

It's all mean. Could it be put at a single place in methods, so we do not have to repeat it 12x in figure legends?

6. Please note that scale bar and its definition are missing for figures 5c-d.

2 mm scale bar was added (is visible in source data).

I would like to suggest some minor changes to the last sentences of the abstract that needs to be written in present tense:

dsRNA expression from a transgene yields sufficient siRNA levels to induce RNAi in heart and muscle. Dicer Δ HEL1/wt mice with enhanced canonical RNAi offer a platform for examining potentials and limits of mammalian RNAi in vivo.

revised

EMBO press papers are accompanied online by A) a short (1-2 sentences) summary of the findings and their significance, B) 2-3 bullet points highlighting key results and C) a synopsis image that is exactly 550 pixels wide and 200-600 pixels high (the height is variable). You can either show a model

or key data in the synopsis image. Please note that text needs to be readable at the final size. Please send us this information along with the final manuscript.

Summary and bullet points were added to the manuscript after the keywords in blue font. A synopsis image was designed.

Dear Petr,

Thank you for the submission of your final ms files. It looks good overall, and thank you also for sending the pdf file of Table EV2, which looks much better than the file our system generated.

The last comments I have regard the info on the statistics that will need to be added to the figure legends, I am afraid.

It is EMBO press policy that the figures are self-explanatory and that the reader does not need to go through other ms files to find relevant information. Can you please address points 2, 3 and 5 and add the missing information where necessary to the figure legends? You can also just add one sentence at the end of the legend and specify n or the mean for specific panels, if this makes it easier. I personally do not find it disruptive to the reading if "n=3 biological replicates" and "data show mean +/- SD" is added to the legends. Point 3 regards the nature of n (I think) so it needs to be added whether n is technical or biological replicates.

You also mention under point 2 that "EV4A and B are just another display of data from Fig 5C and 6B", I think it will be useful to add this information to the EV4 figure legend, thank you.

I also slightly shortened the short summary (we have a word limit) and bullet points. Please let me know whether you agree with this:

Genetic modification of the Dicer gene in mice increases siRNA production from long dsRNA in somatic organs, but silencing by RNAi in vivo is limited by available dsRNA. Mice with enhanced dsRNA levels will enable to study antiviral potential of mammalian RNAi in vivo. (this is still 7 words too long but I think we will be able to get this through)

- Mice tolerate the modification of one allele of Dicer to the highly active Δ HEL1 variant.
- Dicer Δ HEL1/wt mice produce higher levels of mirtrons but affect other miRNAs minimally when a full-length Dicer variant is present.
- Dicer Δ HEL1/wt mice show an order of magnitude higher siRNA production from long dsRNA but efficient RNAi requires siRNA levels comparable to those of highly abundant miRNAs

If these last issues will be clarified, I am very happy to accept your ms. Please send us a new ms file by email and we can upload it for you. I am sorry for this somewhat lengthy process and it is totally fine if you send us all next week.

Best wishes
Esther

Dear Esther, attached is the corrected manuscript, changes in legends were done in blue font. Specific comments on points 2, 3, 5 from data editors are below:

2. Please note that information related to n is missing in the legends of figures 3a-b, g; 4e, supplementary figures 1b; 4a, c.

3a – The MosIR plasmid was transfected into ESCs with indicated genotypes in triplicates in which small RNAs were analyzed by RNA-seq.

3b – “Two or more transfection experiments were performed in a triplicate for each genotype and inverted repeat plasmid.”

3g – (currently 3F) -“from three animals for each genotype”.

4e – now 5E, added n=3

EV1b - Growth curves of males and females on the ICR background. 8 animals were analyzed for each genotype. Data are presented as mean +/- SD error bars.

EV4a,c (currently EV4a,b) – added, both now refer to main figures and highlight that these panels represent a log2 display (which resolves better values around that of Hprt).

3. Although 'n' is provided, please describe the nature of entity for 'n' in the legends of figures 3d-e; 5f.

3d, e n=3 animals for each genotype.

5f (now 6f) n=3 animals for each genotype.

5. Please note that the measure of center for the error bars needs to be defined in the legends of figures 1c;

3a-b, d-e, g; 4c, e-f; 5b, e-f, supplementary figures 4a-c.

Added at the end of each figure legend: All bar graphs depict mean +/- SD error bars.

The different display in EVs is now articulated there as well. My additional comments to the changes in the synopsis are below in blue font.

With kind regards

Petr

Modification of the Dicer gene in mice increases siRNA production in somatic organs, but dsRNA abundance must be strongly increased for achieving efficient RNAi in vivo. Mice with enhanced RNAi will help to evaluate antiviral potential of mammalian RNAi in vivo.

this version is 3 words shorter

Petr Svoboda
Institute of Molecular Genetics of the Czech Academy of Sciences
Laboratory of Epigenetic Regulations
Videnska 1083
Prague 14220
Czech Republic

Dear Petr,

I am very pleased to accept your manuscript for publication in the next available issue of EMBO reports. Thank you for your contribution to our journal.

Best,
Esther
